# Reappraisal of the extinct barbelthroat shark †*Bavariscyllium* and the nebulous origin of carcharhiniform galeomorphs
Sebastian Stumpf [1,2] ✉, Julia Türtscher [2,3], Faviel A. López-Romero [4], Eduardo Villalobos-Segura[2], Arnaud Begat[2,5], Manuel Amadori[2], Richard P. Dearden[6,7], Bruce Lauer [8], René Lauer [8], Andreas Hecker[9] & Jürgen Kriwet [2,5] ✉

We present a revised diagnosis of the extinct galeomorph shark †*Bavariscyllium* based on dental and skeletal material from the Upper Jurassic of Germany and test its purported carcharhiniform affinity through morphometric and phylogenetic analyses. Although †*Bavariscyllium* possesses a whisker-like throat barbel suggesting a closer relationship with orectolobiforms, our findings reveal insufficient evidence to confidently assign †*Bavariscyllium* to either Orectolobiformes or Carcharhiniformes. Additionally, we present quantitative evidence indicating that early galeomorphs, despite probably not being placed among extant orders, were exploring a variety of body forms, predating the divergence of most major body plans among modern representatives. †*Bavariscyllium* exhibits generalised clutching-type teeth with a hemiaulacorhize root characterised by strongly flared root lobes, closely resembling the supposed earliest carcharhiniforms from the Middle Jurassic. However, these features neither confirm nor refute a carcharhiniform affinity, questioning the reliability of these early galeomorphs as calibration fossils for dating the divergence of carcharhiniforms in phylogenomic analyses.

Modern sharks (selachimorph elasmobranchs) are an iconic group of predatory vertebrates that emerged in the Palaeozoic but did not diversify until the early Mesozoic[1–3]. Comprising more than 540 living species[4], modern sharks are divided into two major groups, the Galeomorphii and the Squalomorphii[5–9]. The galeomorph order Carcharhiniformes, with more than 290 extant species in 54 genera, represents the most widespread and species-rich clade of living sharks[4,10], and new species continue to be described (e.g.[11,12]). Commonly known as ground sharks, carcharhiniforms have evolved a diverse array of ecological adaptations that allow them to thrive in a variety of marine habitats, spanning from marginal marine waters to the deep sea and encompassing both temperate and tropical regions around the world[4,13,14]. However, understanding the early evolution of carcharhiniforms (and sharks in general) is complicated because most fossil species are only known from their teeth, which provide important morphological information for species identification and establishing reliable diagnoses[15], but do not necessarily reflect phylogenetic relationships (e.g.[16–18]). Given the poor preservation potential of their cartilaginous endoskeletons, skeletal remains, which offer important clues for inferring phylogenetic relationships and ecomorphological adaptations, are rare and limited to a few localities (e.g.[19–21]).

The carcharhiniform fossil record, as currently understood, dates back to the Middle Jurassic (Bathonian)[15,22], but it was not until well into the Cretaceous that they began to diversify[23,24]. The oldest fossil skeletons proposed to have a carcharhiniform affinity come from the Upper Jurassic of Europe and are represented by three species in three genera. While †*Corysodon cirinensis* is tentatively regarded as Carcharhiniformes *incertae familae*[25], both †*Palaeoscyllium formosum* and †*Bavariscyllium tischlingeri* have been placed among catsharks in the family Scyliorhinidae[22,26,27]. Scyliorhinidae, long considered the largest of all living shark families, traditionally includes approximately 160 extant species of small-bodied sharks

[1]Natural History Museum Vienna, Geological-Palaeontological Department, Vienna, Austria. [2]University of Vienna, Department of Palaeontology, Geozentrum, Vienna, Austria. [3]Johannes Kepler University Linz, Library, Altenberger Straße 69, 4040, Linz, Austria. [4]Universidad Nacional Autónoma de México, Unidad de Sistemas Arrecifales, Instituto de Ciencias del Mar y Limnología, Puerto Morelos, Quintana Roo, México. [5]University of Vienna, Vienna Doctoral School of Ecology and Evolution (VDSEE), Vienna, Austria. [6]Naturalis Biodiversity Center, Vertebrate Evolution, Development, and Ecology, Leiden, The Netherlands. [7]University of Birmingham, School of Geography, Earth & Environmental Sciences, Edgbaston, Birmingham, UK. [8]Lauer Foundation for Paleontology, Science and Education, Wheaton, IL, USA. [9]Jura-Museum Eichstätt, Eichstätt, Germany. ✉e-mail: sebastian.stumpf@nhm.at; juergen.kriwet@univie.ac.at

with a generalised clutching-type dentition and with the first dorsal fin located above or behind the pelvic fins[13,14,16,28,29]. However, both morphological and molecular evidence have consistently rejected the monophyly of Scyliorhinidae *sensu lato*[5,6,9,29–32], thus rendering the higher-level classification of fossil sharks with scyliorhinid-like teeth difficult.

†*Bavariscyllium*, initially described on the basis of a single complete specimen from the Late Jurassic Solnhofen Archipelago in southern Germany[27], is a small shark resembling extant representatives of the leptobenthic ecomorphotype *sensu*[33] (see also[10]). It is distinguished from its contemporaries †*Corysodon* and †*Palaeoscyllium* in its elongate precaudal body design, rounded pectoral and pelvic fins, and a low caudal fin without a ventral lobe. The dorsal fins are also rounded and located far back on the body, with the first dorsal fin originating behind the pelvic fin insertions. The anal fin is low and elongate, extending from the posterior edge of the first dorsal fin to the posterior edge of the second dorsal fin. The dentition of †*Bavariscyllium*, as described based on a few fragmentary teeth extracted from the holotype specimen[27], remains poorly known. Additional dental material includes fragmentary teeth from the Kimmeridgian of northern Germany, assigned to †*Bavariscyllium* sp.[27], and previously undescribed teeth from the lower Kimmeridgian of Mahlstetten in southern Germany.

Thies[27] assigned †*Bavariscyllium* to Scyliorhinidae, based on the position of the first dorsal fin relative to the pelvic fins and the presence of generalised clutching-type teeth without a uvula or apron. Since its initial description, additional specimens of †*Bavariscyllium* from the Solnhofen Archipelago have come to light[20,34,35], but no detailed study of this fossil material has been conducted. Resch and Lauer[35] reported two specimens, both with a single whisker-like barbel protruding ventrally from the head. As noted by Resch and Lauer[35], this feature is also present in other specimens, including the holotype, although it went unrecognised by Thies[27] in his original description. As such, †*Bavariscyllium* appears to be readily distinguished from all living sharks. The latter commonly possess a bilateral pair of barbels associated with their nostrils[4], innervated by cranial nerves[36] and serving as sensory organs for prey detection and environmental sensing through electroreception and possibly mechanoreception[37], although they do not appear to play a role in chemoreception[38]. An exception is the extant bottom-dwelling shark *Cirrhoscyllium*, which comprises three species from the northwestern Pacific and is placed in the galeomorph order Orectolobiformes, in the family Parascylliidae[39,40]. It is the only known shark to possess, in addition to short nasal barbels, a unique pair of whisker-like barbels originating from the ventral surface of the throat, likely serving a mechanosensory function[41]. Parascylliids are considered the sister group to all other extant orectolobiforms[6,9,39] and share with †*Bavariscyllium* features such as a generalised scyliorhinid-like clutching-type dentition[17] and an elongate, fusiform precaudal body design, with the first dorsal fin emerging well behind the origin of the pelvic fins[4,40]. These shared features thus cast doubt on the currently accepted systematic placement of †*Bavariscyllium* and conversely suggest an orectolobiform rather than a carcharhiniform affinity. This study presents: (1) a revised diagnosis of the extinct barbelthroat shark †*Bavariscyllium tischlingeri* based on dental and skeletal material from the Upper Jurassic of southern Germany; (2) an evaluation of its purported carcharhiniform nature through the implementation of both morphometric and phylogenetic analyses; and (3) a discussion of its implications for understanding early galeomorph evolution.

# Results
## Systematic palaeontology
Class **Chondrichthyes** Huxley[42]
Subclass **Elasmobranchii** Bonaparte[43]
Superorder **Galeomorphii** Compagno[44]
*Incerti ordinis*
*Incertae familiae*
†*Bavariscyllium* Thies[27]
**Type species**. †*Bavariscyllium tischlingeri* Thies[27].

**Revised diagnosis**. Small galeomorph shark characterised by the following combination of morphological characters: slender and elongate precaudal body; head short with a whisker-like throat barbel; pectoral fins relatively small, aplesodic and rounded in shape; scapular process pointed with postero-dorsal process at its base; pelvic fins small, aplesodic and broadly angular to rounded, originating at about one-third of the total length; two relatively large dorsal fins with rounded tips lacking spines and basal plates, with the first dorsal fin originating posterior to the pelvic fin insertions; second dorsal fin equal to or slightly larger than the first dorsal fin; interdorsal space slightly wider than the dorsal fins are long; caudal fin low and very long, accounting for about one-third of the total length, with a subterminal notch but no ventral lobe; vertebral centra cyclospondylic; teeth with well-defined main cusp and a pair of lateral heels, each bearing no more than three, usually two, reduced lateral cusplets; labial crown face concave along its basal edge and overhanging the root; apron and uvula absent; cutting edges weak and continuous; labial crown face either devoid of ornamentation or with two oblique ridges extending along the lateral heels from which weak vertical ridges may branch off; well-developed enameloid collar along crown-root junction; low Y-shaped root with flat basal face and strongly flared labial root lobes; root vascularization of hemiaulacorhize type; lingual protuberance of the root pierced by single foramen, connected to an internal canal to central basal foramen; morphological variation passing posteriorly through the dentition involves a reduction in main cusp size and spreading of labial root lobes.

**Temporal and spatial distribution**. Kimmeridgian–Valanginian; Europe (Germany, France).
†*Bavariscyllium tischlingeri* Thies[27].
Figs. 1–5.
1994 Unbenannter Hai.—Frickhinger[45], Fig. 417A.
1996 Unbenannter Hai.—Resch[46], Fig. 1.
v1999 *Phorcynis catulina* Thiollière, 1852—Frickhinger[47], Fig. 162.
v1999 Unbenannter Hai.—Frickhinger[47], Fig. 170.
v2004 *Synechodus* sp.—Kriwet and Klug[48], Fig. 6D.
v*2005 *Bavariscyllium tischlingeri* Thies, 2005—Thies[27], Figs. 1, 2.
2005 *Bavariscyllium* sp.—Thies[27], Fig. 3.
2007 *Bavariscyllium tischlingeri* Thies, 2005—Thies et al.[49], Fig. 3K.
v2008 *Synechodus* sp.—Klug and Kriwet[50], Fig. 3b.
v2010 *Synechodus* sp.—Klug[51], Fig. 1D.
v2011 *Bavariscyllium tischlingeri* Thies, 2005—Thies and Leidner[52], pls. 44, 45.
v2012 *Bavariscyllium tischlingeri* Thies, 2005—Cappetta[15], Figs. 253, 254.
2014 *Thiesus concavus* Guinot et al., 2014—Guinot et al.[53], Fig. 9P–D‘.
v2015 *Bavariscyllium tischlingeri* Thies, 2005—Kriwet and Klug[34], Fig. 696.
v2019 *Bavariscyllium tischlingeri* Thies, 2005—Resch and Lauer[35], Figs. 1–7.
v2023 *Bavariscyllium tischlingeri* Thies, 2005—Villalobos-Segura et al.[20], Fig. 23.
v2024 *Bavariscyllium tischlingeri* Thies, 2005—Duffin and Batchelor[54], Fig. 11G–I.

**Diagnosis**. Same as for genus (by monotypy).

**Holotype**. JME SOS 4124, a complete articulated dentition-bearing specimen.

**Type locality and age**. Eichstätt, Bavaria, Germany; *Hybonoticeras hybonotum* ammonite zone, early Tithonian, Altmühltal Formation[55,56].

**Referred specimens**. Five articulated specimens from the Late Jurassic Solnhofen Archipelago in Bavaria, southern Germany (JME SOS 4124 from the lower Tithonian of Eichstätt; LF 1436 from the lower Tithonian of Eichstätt; SMF P 272 from the lower Tithonian of Eichstätt; SMNS 96086 from the lower Tithonian of Eichstätt; SNSB-BSPG 1878 IV 6 from the lower Tithonian of Solnhofen); and 17 isolated teeth from the lower Kimmeridgian of Mahlstetten in Baden-Württemberg, southern Germany (SMNS 89612/1–17).

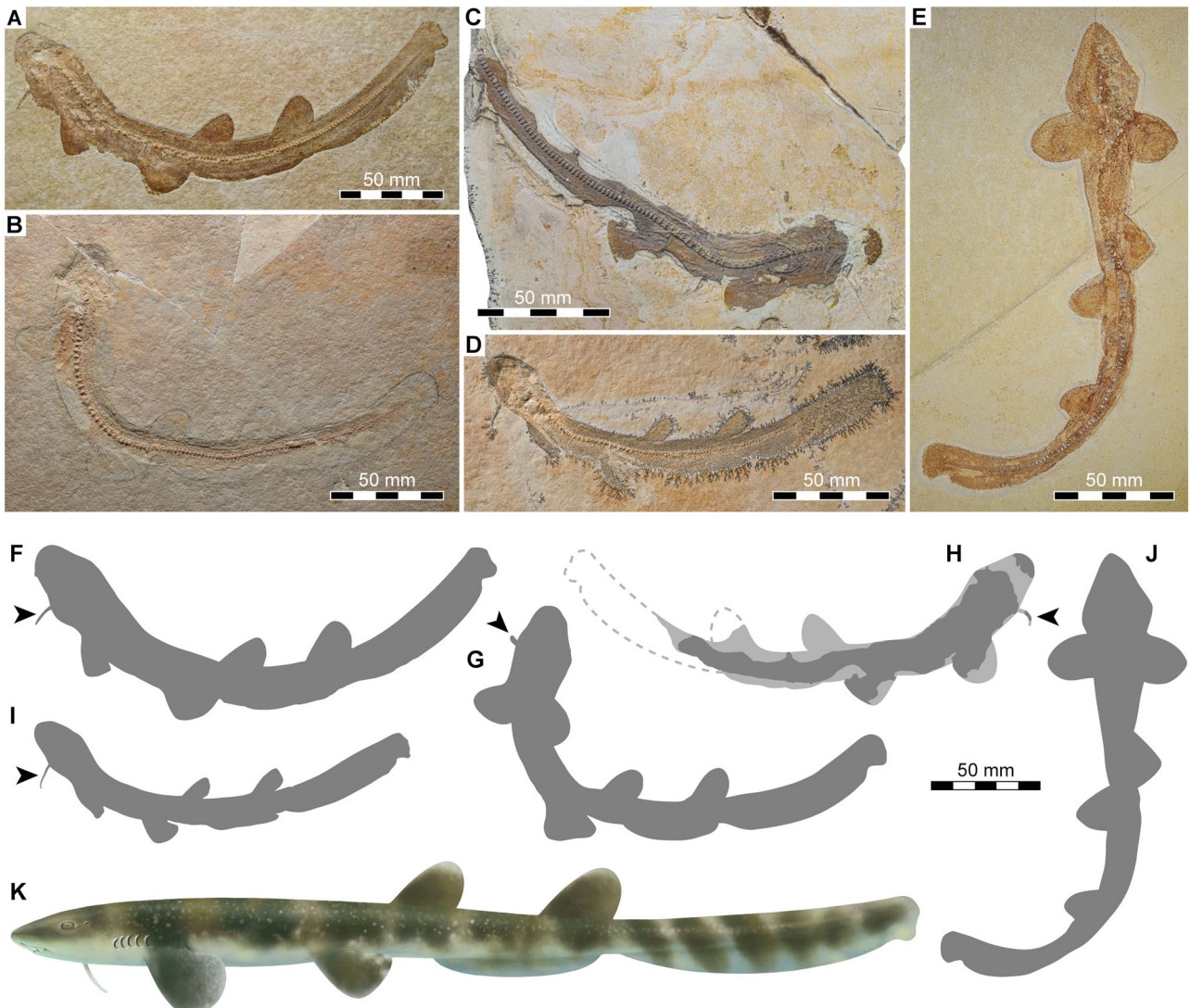

**Fig. 1 | Articulated skeletons of †*Bavariscyllium tischlingeri* from the Late Jurassic Solnhofen Archipelago in southern Germany. A, F** JME SOS 4124 (holotype) from the lower Tithonian of Eichstätt; **B, G** SMF P 272 from the lower Tithonian of Eichstätt; **C, H** SNSB-BSPG 1878 IV 6 from the lower Tithonian of Solnhofen; **D, I** LF 1436 from the lower Tithonian of Eichstätt; **E, J** SMNS 96086 from the lower Tithonian of Eichstätt (arrows indicate throat barbel); **K** artistic life reconstruction of †*Bavariscyllium tischlingeri*.

## Description

**External morphology and endoskeleton.** The skeletons from the Late Jurassic Solnhofen Archipelago are articulated, with densely arranged dermal denticles outlining the body and fins (Fig. 1), providing limited information beyond external morphology (Figs. 2–4). The body of †*Bavariscyllium* is elongate and slender, with a total length ranging from approximately 190 mm (SMF P 272) to 250 mm (SMNS 96086), reflecting a variation of 24% in total length between the smallest and largest specimen. The body appears to have been slightly laterally compressed, as most specimens (including those in private collections[35,45,46]) are preserved in lateral view.

The endoskeleton, where discernible, comprises elements of both the axial and appendicular skeleton. The endoskeleton of SMF P 272, while appearing fully developed (Fig. 3C–E), seems less well-mineralised than that of the larger specimens, suggesting an earlier developmental stage. Despite the small size of the available specimens and the absence of claspers in any of them, which might suggest juvenile stages, they are interpreted as representing subadult to adult individuals. This interpretation is based on the apparently advanced mineralization of both their endo- and exoskeletons, in conjunction with the prolonged skeletogenesis observed in extant sharks, which starts during late embryonic development and continues well beyond hatching (e.g.[57–59]), with males usually developing well-mineralised claspers during late ontogeny, upon attaining sexual maturity (e.g.[60–62]).

The head is short, terminating in a bell-shaped snout. A single whisker-like barbel protrudes ventrally from the throat at approximately the level of the jaw articulation (Figs. 1A–D, F–I and 2A, B, E–G). This feature is consistent across almost all known specimens of †*Bavariscyllium* (including those in private collections[35,46]), with the exception of SMNS 96086 (Fig. 1E, J), but this is likely to be a preservational artefact resulting from the dorsoventral compression of the specimen. Nevertheless, the possibility that †*Bavariscyllium* may have possessed two throat barbels cannot be ruled out entirely. Consequently, the presence of a single throat barbel is here regarded as tentative unless substantiated by further fossil evidence. Additionally, a small flap of dermal denticles protruding antero-laterally from the snout of JME SOS 4124 (Fig. 2A, B) is here tentatively identified as the left nasal barbel. The neurocranium and splanchnocranium are visible in SNSB-BSPG 1878 IV 6, although both are incomplete and obscured by squamation, preventing the detailed description of individual elements (Fig. 3A, B). The Meckel's cartilage and ceratohyal appear to have been elongate and rather gracile, but due to the incomplete preservation,

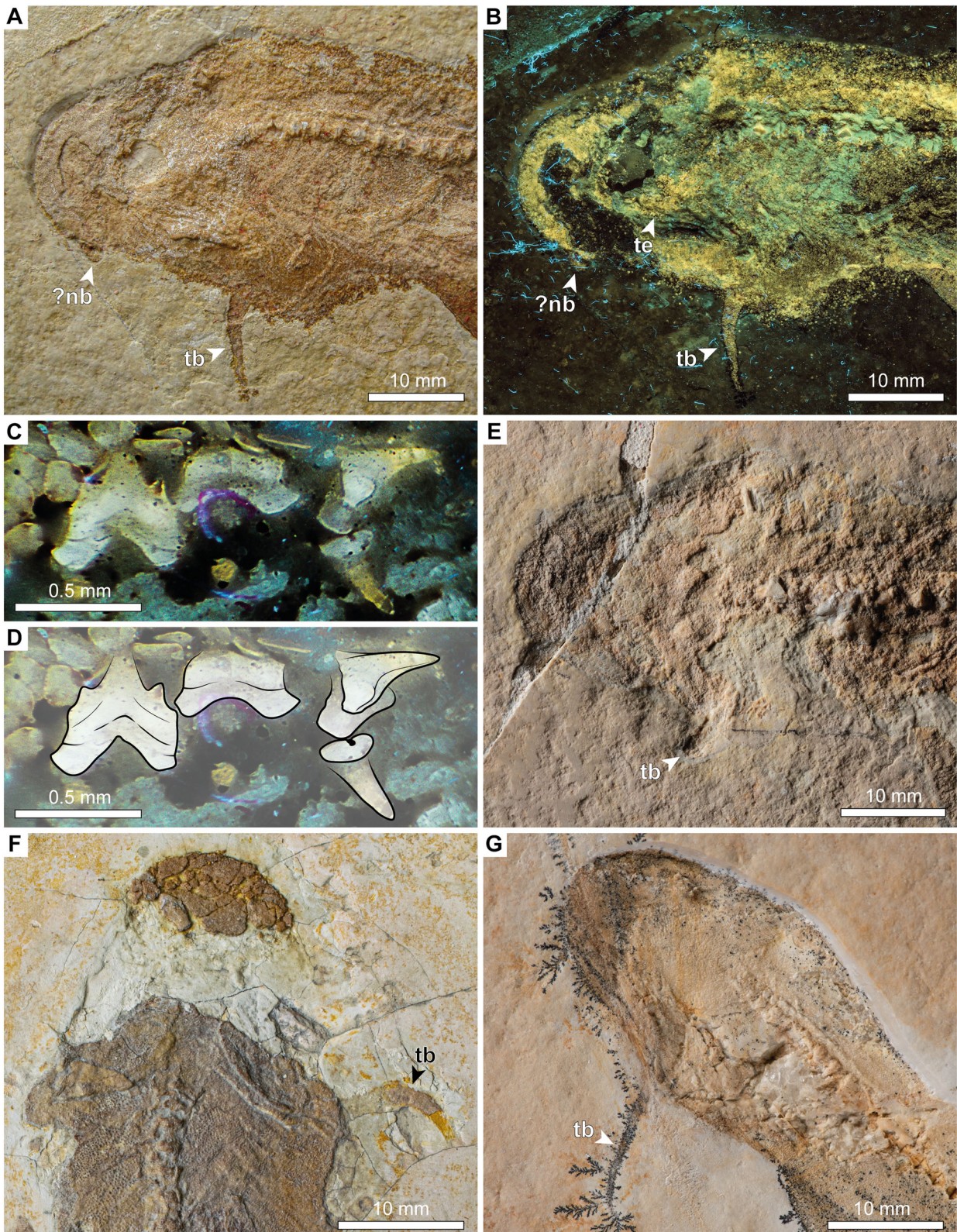

**Fig. 2 | Head of †*Bavariscyllium tischlingeri* from the Late Jurassic Solnhofen Archipelago in southern Germany. A**, **B** JME SOS 4124 (holotype) under **A** normal light and **B** UV light, captured with a yellow filter; **C**, **D** close-ups of lateral teeth of JME SOS 4124 under UV light; **E** SMF P 272; **F** SNSB-BSPG 1878 IV 6; **G** LF 1436. Anatomical abbreviations: nb nasal barbel, te teeth, tb throat barbel.

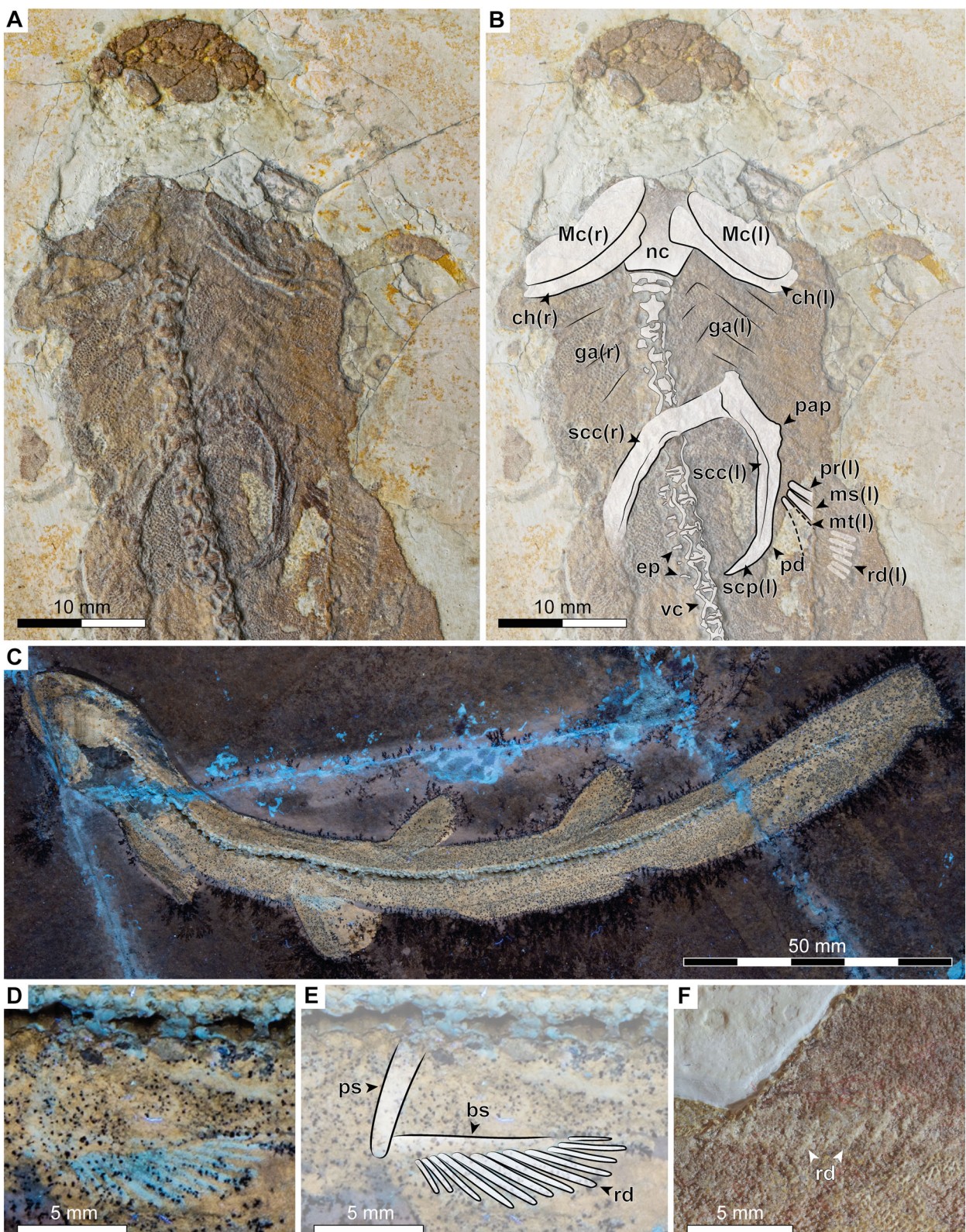

**Fig. 3 | Endoskeleton of †*Bavariscyllium tischlingeri* from the Late Jurassic Solnhofen Archipelago in southern Germany. A, B** Head and pectoral region of SNSB-BSPG 1878 IV 6; **C** LF 1436 under UV light; **D, E** pelvic girdle of LF 1436 under UV light; **F** first dorsal fin base of JME SOS 4124 (holotype). Anatomical abbreviations: bs basipterygium, ch ceratohyal, ga gill arches, ep epaxial elements of vertebra, Mc Meckel's cartilage, ms mesopterygium, mt metapterygium, nc neurocranium, pap articular process for pectoral fin, pd postero-dorsal process at base of scapular process, pr propterygium, ps puboischiadic bar, rd radial, scc scapulocoracoid, scp scapular process, vc vertebral centrum.

their precise morphology cannot be determined. The gill arches are barely discernible, preventing any meaningful morphological insights.

The pectoral fins are relatively small, aplesodic, and rounded in shape. The pectoral girdle is best preserved in SNSB-BSPG 1878 IV. The paired scapulocoracoids are fused ventrally at their coracoid portions (Fig. 3A, B). The process for the articulation with the pectoral fin is moderately well-developed, but no foramina for the diazonal nerves could be observed. The scapula ends distally in a pointed scapular process with a weakly developed postero-dorsal process at its base. The pectoral fin is supported by three basal cartilages, the propterygium, mesopterygium, and metapterygium. Although these cartilages are incompletely preserved and partially obscured by squamation, they all appear to be elongate and straight. Furthermore, at least six pectoral radials are discernible in SNSB-BSPG 1878 IV, but due to their obscured nature, no further details can be provided. The pelvic fins are located anteriorly on the body, originating at about one-third of the total length. They are small, aplesodic and broadly angular to rounded. The pelvic girdle is barely discernible in any of the specimens. The puboischiadic bar seems to have been straight lacking any processes (Fig. 3D, E). Similarly, the basipterygium appears to have been straight, supporting at least 12 pectoral radials. The dorsal fins are located back on the body, with the first dorsal fin originating posterior to the pelvic fin insertions. They are relatively large and roughly triangular with rounded tips, lacking spines and basal plates, and are supported by a single set of radials (Fig. 3F). The second dorsal fin is either equal to or slightly larger than the first dorsal fin. The interdorsal space is slightly wider than the dorsal fins are long (Fig. 1). The caudal fin is low and elongate, with a distinct subterminal notch but no ventral lobe. It accounts for approximately one-third of the total length and originates just behind the second dorsal fin. There is no upper or lower caudal crest of dermal denticles. The anal fin is low and elongate, originating below or just behind the first dorsal fin insertion and terminating below the second dorsal fin insertion.

The vertebral column is best preserved in JME SOS 4124, which comprises at least 115 cyclospondylic vertebral centra (Fig. 4); however, insufficient preservation of the posterior-most centra precludes a more precise count. The transition from mono- to diplospondylous vertebrae occurs above the pectoral fins (Fig. 4B), marked by a distinct decrease in anterior-posterior length between vertebrae 28 and 29 (Fig. 4C). A progressive decrease in vertebral length takes place along the caudal fin towards its posterior end. Epaxial (dorsal) elements are discernible, likely representing basidorsals and interdorsals, although a more precise determination is limited due to poor preservation. Hypaxial (ventral) elements could not be observed, but this might be an artefact of preservation.

**Dentition**. The dentition of †*Bavariscyllium tischlingeri* remains poorly understood, with previous information limited to four incomplete teeth extracted from the holotype specimen, which Thies[27] identified as one anterior, one antero-lateral, one lateral, and one posterior tooth, all missing most of their roots. The exact origin of these teeth, whether from the lower or upper dentition, remains undetermined.

Our re-examination of JME SOS 4124 revealed the presence of four reasonably well-preserved lateral teeth, including one that is nearly complete (Fig. 2C, D), which appear to have been overlooked in the original description by Thies[27]. However, JME SOS 4124 is coated with an unknown transparent consolidant that is permeable to UV light, presumably applied to seal and stabilise the specimen and containing numerous embedded dust particles and fibres. This coating impedes the visibility of the teeth under normal light conditions, although they become considerably more discernible under UV light illumination. Nevertheless, the consolidant obscures minute morphological features, such as crown ornamentation. Despite these limitations, the overall morphology of the preserved teeth remains sufficiently identifiable to permit meaningful morphological assessment. As with the extracted teeth described by Thies[27], the assignment of these additional teeth to either the lower or upper dentition remains uncertain. The teeth are visible in labial, lingual, and lateral views, and

exhibit a pointed main cusp flanked by one or two pairs of small lateral cusplets. Notably, the lateral cusplets appear slightly more developed than those in the extracted teeth described by Thies[27], indicating a higher degree of heterodonty than previously recognised. Continuous cutting edges are present, running across the main cusp and lateral cusplets. Tooth crown ornamentation appears to be reduced and includes at least two faint oblique ridges extending along the labial crown face. The labial crown face exhibits a slightly concave basal edge, and the root is bilobate, with diverging root lobes protruding labially below the crown. The lingual protuberance of the root appears moderately well-developed and is basally pierced by a single foramen.

The isolated teeth from the lower Kimmeridgian of Mahlstetten in southern Germany, which share key dental features with the holotype, provide additional morphological insights that help clarify the dental morphology of †*Bavariscyllium tischlingeri*.

The teeth from Mahlstetten are very small, measuring less than 1 mm in height, and can be separated into those coming from tooth files of antero-lateral, lateral, and posterior positions (Fig. 5). Anterior teeth, which are the largest in the jaws and characterised by a very high and slender main cusp[27], could not be identified. The teeth are generally well-preserved, but have suffered post-mortem damage in some places, with some teeth displaying patterns of bioerosion, likely caused by endolithic microorganisms (compare[63]).

Antero-lateral teeth (Fig. 5A–E) are slightly asymmetrical and higher than wide, with a high, pointed main cusp. Below the main cusp, the crown slightly widens, giving rise to a pair of distally diverging heels and resulting in a labial crown face with a concave basal edge that overhangs the root (Fig. 5A). The lateral heels are labially displaced with respect to the main cusp, resulting in a flat or slightly convex labial crown face. Each heel bears two pairs of rudimentary cusplets, the most mesial and distal of which are barely discernible. There is neither an apron nor a uvula. The cutting edges are weak and continuous across the main cusp and lateral cusplets (Fig. 5B). The labial crown face bears two oblique ridges that extend along the lateral heels up to the base of the main cusp. Additionally, a few straight to slightly undulating vertical ridges run along the main cusp and lateral heels. The lingual crown face exhibits a few vertical ridges that are confined to the lower half of the main cusp (Fig. 5C). There is a well-developed enameloid collar that covers the uppermost part of the root along the entire crown-root junction.

The root is very low and flat with a flared basal face (Fig. 5D). It is Y-shaped in basal view with a low, slightly swollen lingual protuberance and flared root lobes that protrude below the lateral heels of the crown. The root vascularization is hemiaulacorhize, with the lingual protuberance of the root pierced basally by a single foramen that connects to the central basal foramen by an internal canal (Fig. 5C). Laterally, at the level of the first pair of lateral cusplets immediately below the collar, there is a single large foramen, accompanied by a series of smaller foramina extending below the second pair of lateral cusplets (Fig. 5B, E).

Lateral teeth (Fig. 5F–P), similar to antero-laterals, are higher than wide but symmetrical, with more well-developed lateral heels and cusplets, and more strongly flared labial root lobes that extend well below the lateral heels of the crown (Fig. 5G, J, N). The labial crown face has a strongly concave basal edge and may be devoid of ornamentation (Fig. 5N), but it usually exhibits two oblique ridges that ascend along the lateral heels up to the lower part of the main cusp (Fig. 5G, J). From these ridges, weak vertical ridges may branch off, and straight to slightly undulating vertical ridges may occur on the main cusp, but do not reach its apex. The vertical ridges extending lingually may reach the apex of the main cusp (Fig. 5L, P). As seen in antero-lateral teeth, a single large foramen and an accompanying series smaller foramen may be present, penetrating the root laterally immediately below the lateral cusplets (Fig. 5H, I). Labially, the root may be perforated by a row of foramina below the collar where the lateral root lobes join (Fig. 5J, M).

Lateral teeth from more posterior positions (Fig. 5R–U) are asymmetrical, displaying a distally inclined main cusp and a slightly more

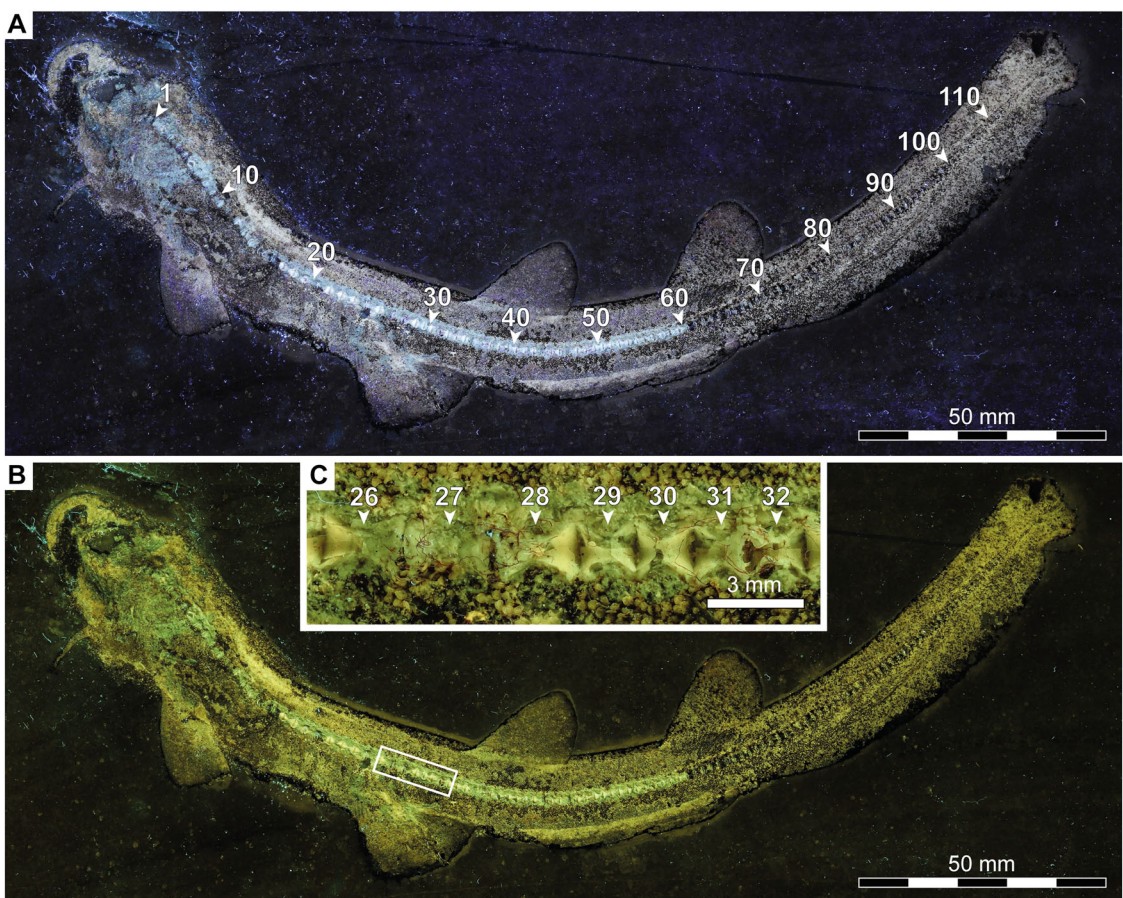

**Fig. 4 | Vertebral column of †*Bavariscyllium tischlingeri* from the Late Jurassic Solnhofen Archipelago in southern Germany. A–C** JME SOS 4124 (holotype) under UV light, captured without (**A**) and with (**B, C**) a yellow filter. Numbers indicate individual vertebrae.

elongate mesial crown heel and root lobe (Fig. 5R, T). The lateral cusplets are reduced and may be barely discernible (Fig. 5R).

Exhibiting the characteristic ornamentation of lateral teeth, posterior teeth (Fig. 5V–C′) are about as high as wide, with a short but stout main cusp and typically up to two pairs of lateral cusplets. The mesial heel may bear three pairs of lateral cusplets, the most mesial one being very minute and barely visible (Fig. 5X). No foramina perforating the root could be observed.

**Dermal denticles (squamation).** The dermal denticle morphology of †*Bavariscyllium tischlingeri* has been documented by Thies and Leidner[52], who figured five isolated dermal denticles from the holotype specimen (Thies and Leidner[52]: Fig. 44B–E). However, the specific regions of the body from which these denticles originate remain unknown. The crown is leaf-shaped, lacking a median keel, and features a few subtle notches along its anterior margin. Additionally, faint longitudinal ridges are present but do not extend to the posterior margin of the crown.

**Morphometrics**

To quantify the body shape of †*Bavariscyllium tischlingeri*, a set of 16 linear body measurements was used (Supplementary Fig. 1) to compare this species, along with †*Palaeoscyllium formosum*, to extant demersal sharks with elongate, fusiform precaudal body designs, comprising species in the carcharhiniform families Scyliorhinidae, Atelomycteridae and Pentanchidae, and phenotypically similar sharks in the orectolobiform families Parascylliidae and Hemiscylliidae (Fig. 6A). The compiled dataset was subjected to a principal component analysis (PCA), resulting in 15 axes (Supplementary Table 1). The first three axes each explain more than 5% of the variation and together account for 89.51% of the total variability (see Supplementary Table 2 for information on all PC axes, and Supplementary Table 2 for loadings of the first three PC axes). Plotted on PC1 (55.04% of total variation) and PC2 (24.3%), the morphospace occupations shown in Fig. 6B display hemiscylliids distinctly separated from all other taxonomic groups, occupying negative PC1 and primarily positive PC2 values. Pentanchids are widely scattered and occupy the largest morphospace within the dataset, spanning both positive and negative values of PC1 and PC2, although they tend to cluster around positive PC1 and negative PC2 values. There is considerable overlap between pentanchids and scyliorhinids, with the latter predominantly occupying negative values along PC2, showing only a narrow extension into positive PC2 values, and spanning both positive and negative values along PC1. Parascylliids partially overlap with both pentanchids and atelomycterids, and are mostly restricted to negative values along both PC1 and PC2. Atelomycterids exhibit a partial overlap with pentanchids, scyliorhinids, and parascylliids, occupying predominantly negative values along both axes. †*Bavariscyllium* (*n* = four specimens) and †*Palaeoscyllium* (*n* = two specimens) are distinct from one another, with †*Bavariscyllium* occupying a unique morphospace that does not overlap with any extant family. One specimen of †*Palaeoscyllium* overlaps with parascylliids, atelomycterids, and pentanchids, while the other falls outside the morphospace of any living family. Grouping extant species at order level reveals an overlap of †*Palaeoscyllium* with both orectolobiforms and carcharhiniforms (Fig. 6C). By contrast, and despite a maximum size difference of 24% between the smallest and largest specimen of †*Bavariscyllium*, the taxon displays a distinct aggregation, positioned closer to carcharhiniforms (more precisely pentanchids)

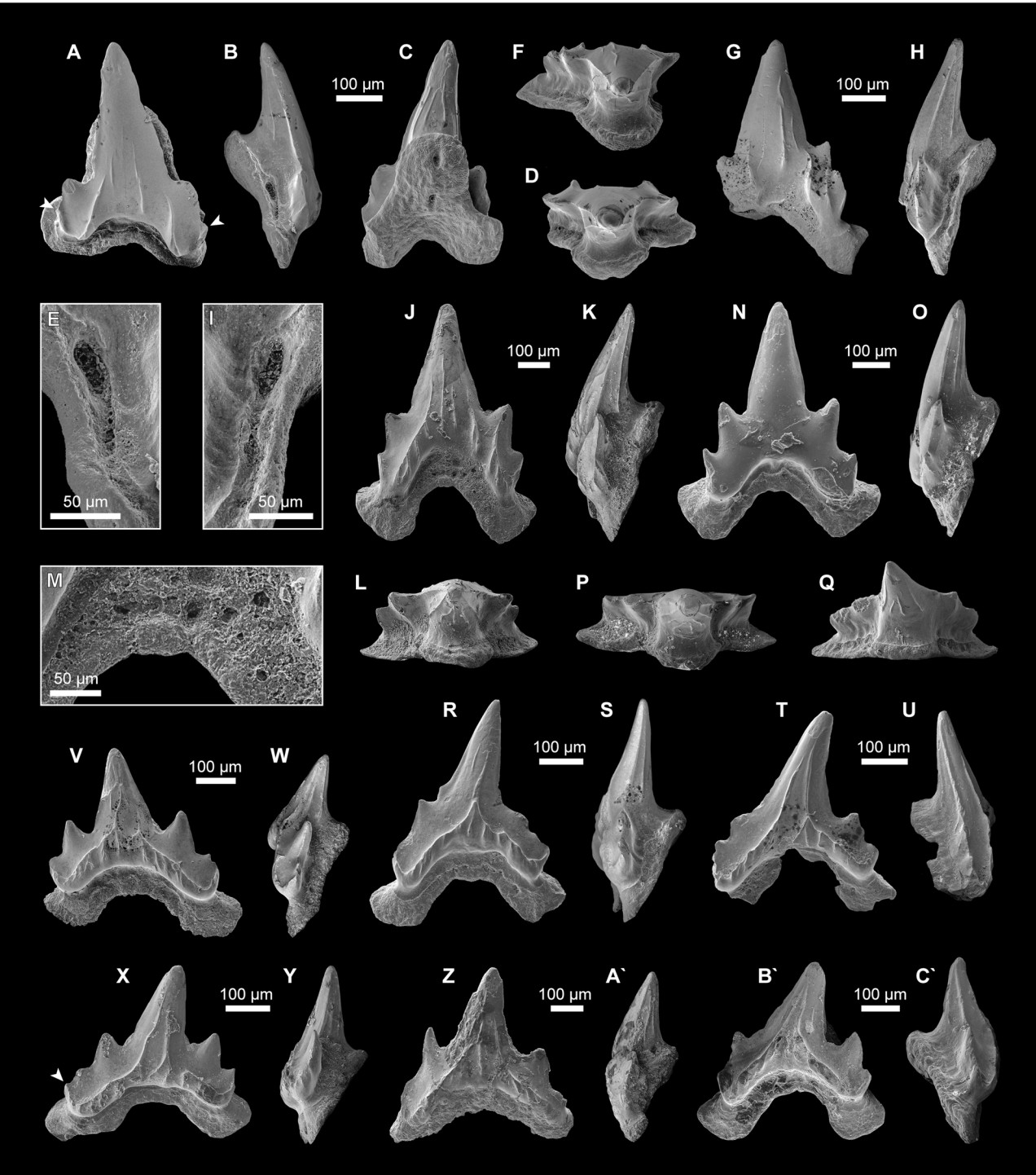

**Fig. 5 | Teeth of †*Bavariscyllium tischlingeri* from the lower Kimmeridgian of Mahlstetten in southern Germany.** A–E Antero-lateral tooth (SMNS 89612/1) in (**A**) labial, (**B**) distal, (**C**) lingual, and (**D**) apical views, with (**E**) a close-up view of the lateral foramen and accompanying foramina penetrating the root; **F–I** lateral tooth (SMNS 89612/2) in (**F**) apical, (**G**) labial, and (**H**) profile views, with (**I**) a close-up view of the lateral foramen and accompanying foramina penetrating the root; **J–M** lateral tooth (SMNS 89612/3) in (**J**) labial, (**K**) profile, and (**L**) apical views, with (**M**) a close-up view of the labial foramina penetrating the root; **N–P** lateral tooth (SMNS 89612/4) in (**N**) labial, (**O**) profile, and (**P**) apical views; **Q–S** postero-lateral tooth (SMNS 89612/5) in (**Q**) apico-lingual, (**R**) labial, and (**S**) profile views; **T, U** postero-lateral tooth (SMNS 89612/6) in (**T**) labial and (**U**) profile views; **V, W** posterior tooth (SMNS 89612/7) in (**V**) labial and (**W**) profile views; **X, Y** posterior tooth (SMNS 89612/8) in (**X**) labial and (**Y**) profile views; **Z, A`** posterior tooth (SMNS 89612/9) in (**Z**) labial and (**A`**) profile views; **B`, C`** posterior tooth (SMNS 89612/10) in (**B`**) labial and (**C`**) profile views. Arrows indicate very weakly developed lateral cusplets.

than to orectolobiforms. This distinct morphospace occupation suggests that overall body shape of †*Bavariscyllium* may have been conserved throughout ontogeny, although this interpretation requires further testing with a larger sample size.

Plotting PC1 against PC3 (10.17% of total variation) reveals a similar morphospace occupation (Supplementary Fig. 3). Hemiscylliids are distinctly separated, occupying exclusively negative values along both axes. Pentanchids exhibit the largest morphospace occupation, spanning both

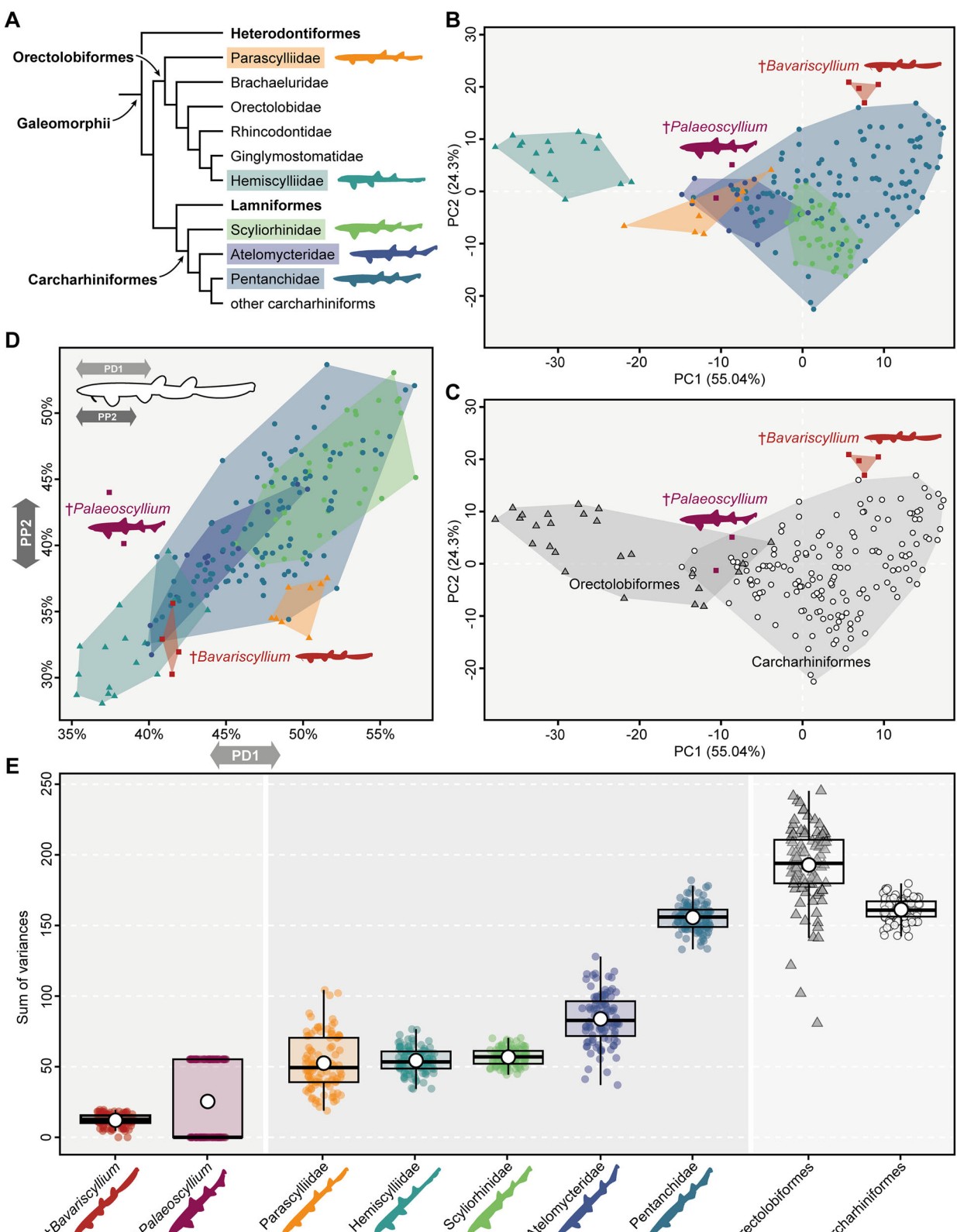

**Fig. 6 | Morphometrics. A** Molecular-based phylogeny of galeomorph sharks (based on Marion et al.[9]) highlighting families included in the morphometric analysis; **B**, **C** results of principal component analysis (PCA) across groups, including four †*Bavariscyllium tischlingeri* and two †*Palaeoscyllium formosum* specimens, with extant species aggregated at (**B**) family (Parascylliidae [*n* = 8]; Hemiscylliidae [*n* = 17]; Scyliorhinidae [*n* = 36]; Atelomycteridae [*n* = 13]; Pentanchidae [*n* = 86]) and (**C**) order level (Orectolobiformes [*n* = 25];

Carcharhiniformes [*n* = 160]); **D** binary plot illustrating the relationship between pre-pelvic fin length (PP2) and pre-first dorsal fin length (PD1), showing extant species aggregated at family level; **E** boxplots illustrating morphological disparity across groups based on 100 bootstrap iterations, with extant species aggregated at family and order level. A large white dot at the centre of each box indicates the mean, and the horizontal line represents the median. Silhouettes of living taxa shown in A and E are based on Ebert et al.[4].

positive and negative values of PC1 and PC3. Scyliorhinids nearly completely overlap with pentanchids, occupying both positive and negative values of PC3 and predominantly positive values of PC1. Parascylliids partially overlap with pentanchids and atelomycterids, restricted to negative PC1 and positive PC3 values. Atelomycterids extend across both positive and negative PC3 values and are almost exclusively negative along PC1, showing partial overlap with parascylliids and pentanchids, and minor overlap with scyliorhinids. †*Bavariscyllium* and †*Palaeoscyllium* are distinct. †*Bavariscyllium* overlaps with pentanchids in the positive regions of PC1 and PC3, while one specimen of †*Palaeoscyllium* overlaps with parascylliids and atelomycterids in the negative region of PC1 and positive region of PC3. The second †*Palaeoscyllium* specimen plots near Parascylliidae.

The relatively wide separation between the two †*Palaeoscyllium* specimens is intriguing, but given the substantial size difference between them (Supplementary Fig. 2), it may be attributed to ontogenetic allometry, as has been documented in extant sharks (e.g.[64–66]). One of the most striking feature distinguishing †*Palaeoscyllium* from †*Bavariscyllium* (and any other taxon included in the present dataset) is the position of the first dorsal fin, which originates anterior to the pelvic fins (Fig. 6D). This condition, along with a slightly more robust precaudal body and a truly heterocercal caudal fin with a distinct ventral lobe, indicates a closer resemblance to †*Corysodon*[25,67] and thus to sharks of the littoral rather than the leptobenthic ecomorphotype[10,33].

A Shapiro-Wilk test showed that 46.67% of all measurements are normally distributed (i.e., seven out of 15 measurements; Supplementary Table 3). A Kruskal–Wallis test revealed significant differences at family level for all non-normally distributed measurements (Supplementary Table 4) and significant differences at order level for all but one non-normally distributed measurement (i.e., interdorsal space [IDS]; Supplementary Table 8). Significant differences were also found at both family and order levels when pairwise Wilcoxon tests were used (Supplementary Tables 5 and 9). ANOVA tests on each normally distributed measurement likewise demonstrated significant differences at family and order levels (Supplementary Tables 6 and 10). These findings also are corroborated by the results of the pairwise comparisons (Supplementary Tables 7 and 11 for comparisons at family and order levels, respectively).

The dataset was further subjected to a linear discriminant analysis (LDA) to explore the separation between the a priori groups using the leave-one-out cross-validation (LOOCV) approach. The results indicate that neither †*Bavariscyllium* nor †*Palaeoscyllium* can be phenotypically allied with any of the analysed families (Supplementary Table 12), with a LOOCV accuracy of 93.2%. When extant species are grouped at order level, it becomes evident that †*Bavariscyllium* is distinct from both carcharhiniforms and orectolobiforms, with a LOOCV accuracy of 96.3%, while †*Palaeoscyllium* is found to be allied with carcharhiniforms (Supplementary Table 13). A similar picture emerges when the morphological disparity, calculated as the sum of variances, is considered, further highlighting the distinct nature of †*Bavariscyllium* (Fig. 6E; Supplementary Tables 14 and 15). The wide disparity displayed by †*Palaeoscyllium* aligns with the results from the PCA, but a larger sample size is needed to draw any conclusions. Overall, †*Bavariscyllium* exhibits a variance distinct from all comparative groups. Disparity calculated as the sum of ranges yields very similar results to those from the sum of variances (Supplementary Tables 16 and 17). Pairwise comparisons at order level remain consistent, while the family-level subsets show slight changes, particularly with differences between †*Bavariscyllium* and †*Palaeoscyllium*.

### Phylogenetic relationships

To explore the phylogenetic relationships of both †*Bavariscyllium* and †*Palaeoscyllium*, we performed a maximum parsimony analysis using a modified character-taxon matrix of Vullo et al.[8]. The analysis yielded 1944 most parsimonious trees (MPTs), each of 632 evolutionary steps, with a consistency index (CI) of 0.51 and a retention index (RI) of 1. The strict consensus tree (SCT) generated from these trees places both †*Bavariscyllium* and †*Palaeoscyllium* within Galeomorphii, in an unresolved

polytomy with extant carcharhiniforms, orectolobiforms and lamniforms (Fig. 7A; for character reconstructions, see Supplementary Information). This poor resolution is likely due to the high amount of missing data for both †*Bavariscyllium* and †*Palaeoscyllium* (52% and 62%, respectively), preventing a more accurate determination of their phylogenetic relationships. In contrast, the majority rule consensus tree (MRCT) places †*Bavariscyllium* and †*Palaeoscyllium* in a polytomy with carcharhiniforms and lamniforms (Fig. 7B). Excluding †*Palaeoscyllium* from the parsimony analysis resulted in 108 MPTs of 632 steps, with a CI of 0.5, and a RI of 1. The SCT recovers †*Bavariscyllium* in an unresolved relationship with carcharhiniforms, although with weak node support (Fig. 7C).

### Discussion

The European Late Jurassic fossil record, renowned for having yielded a multitude of whole-bodied specimens of sharks and rays from some of the most productive Mesozoic fossil sites in the world[20], offers a unique glimpse into the time when modern sharks and rays were just beginning to diversify[1–3]. However, despite considerable research efforts in recent decades, only a limited number of species have been studied through phylogenetic and/or morphometric analyses up to now[68–74].

Our analyses indicate that there is currently insufficient evidence to confidently assign †*Bavariscyllium* to either Orectolobiformes or Carcharhiniformes. The presence of a single whisker-like throat barbel may represent an autapomorphy unique to †*Bavariscyllium*, potentially suggesting a closer relationship with parascylliids, and consequently with orectolobiforms rather than carcharhiniforms. Given that heterodontiform sharks are considered the sister group to orectolobiforms plus all other galeomorphs[6,7,9], the presence of whisker-like throat barbels, whether singular (as inferred in †*Bavariscyllium*) or paired (as in *Cirrhoscyllium*), may represent the plesiomorphic condition for all galeomorphs more derived than heterodontiforms.

The Albian–Lutetian shark †*Pararhincodon* has recently been shown to be a stem-group member of Parascylliidae[75], comprising five species, the oldest of which is †*P. lehmani*, known from an incomplete and poorly preserved articulated specimen from the Cenomanian of Lebanon[76]. Notably, the holotype specimen of †*P. lehmani* also exhibits a single whisker-like throat barbel (Supplementary Fig. 7). This feature has previously been overlooked, although it remains possible that, like *Cirrhoscyllium*, †*Pararhincodon* may have had two throat barbels, pending the discovery of more complete fossil material.

Unlike living orectolobiforms and †*Pararhincodon*, which share a curved mesopterygium with a concave trailing edge creating a distinct interbasal space between the meso- and metapterygium[39,75,77], the mesopterygium of †*Bavariscyllium* appears to have been straight rather than curved. Extant parascylliids differ from †*Bavariscyllium* and †*Pararhincodon* in having the pro- and mesopterygium fused into a single large cartilage[39,75] and well-defined prepelvic processes on the pelvic girdle[39], although the latter feature remains unknown in †*Pararhincodon*.

†*Pararhincodon* and extant parascylliids differ from more derived orectolobiforms in having scyliorhinid-like teeth lacking an apron or uvula, similar to †*Bavariscyllium*, but with less well-developed root lobes[15,17]. However, the recently described species †*Pararhincodon torquis* from the Upper Cretaceous of the UK provides some evidence of a small labial apron in the recently described species, suggesting that the presence of an apron might represent a plesiomorphic feature rather than a dental trait shared by orectolobiforms more derived than parascylliids[75]. In †*Pararhincodon*, the tooth roots are either hemiaulacorhize or holaulacorhize, depending on the species[15], while in extant parascylliids they always are hemiaulacorhize[17].

The teeth from the lower Kimmeridgian of Mahlstetten, which share characteristic dental features with the holotype of †*B. tischlingeri* (including a concave labial crown edge and a pair of labially displaced lateral heels, each with small cusplets and typically an oblique labial ridge, as well as divergent labial root lobes and a weak lingual protuberance pierced basally by a single foramen), provide valuable insights into the species' dental morphology, which has not been described previously. These scyliorhinid-like teeth,

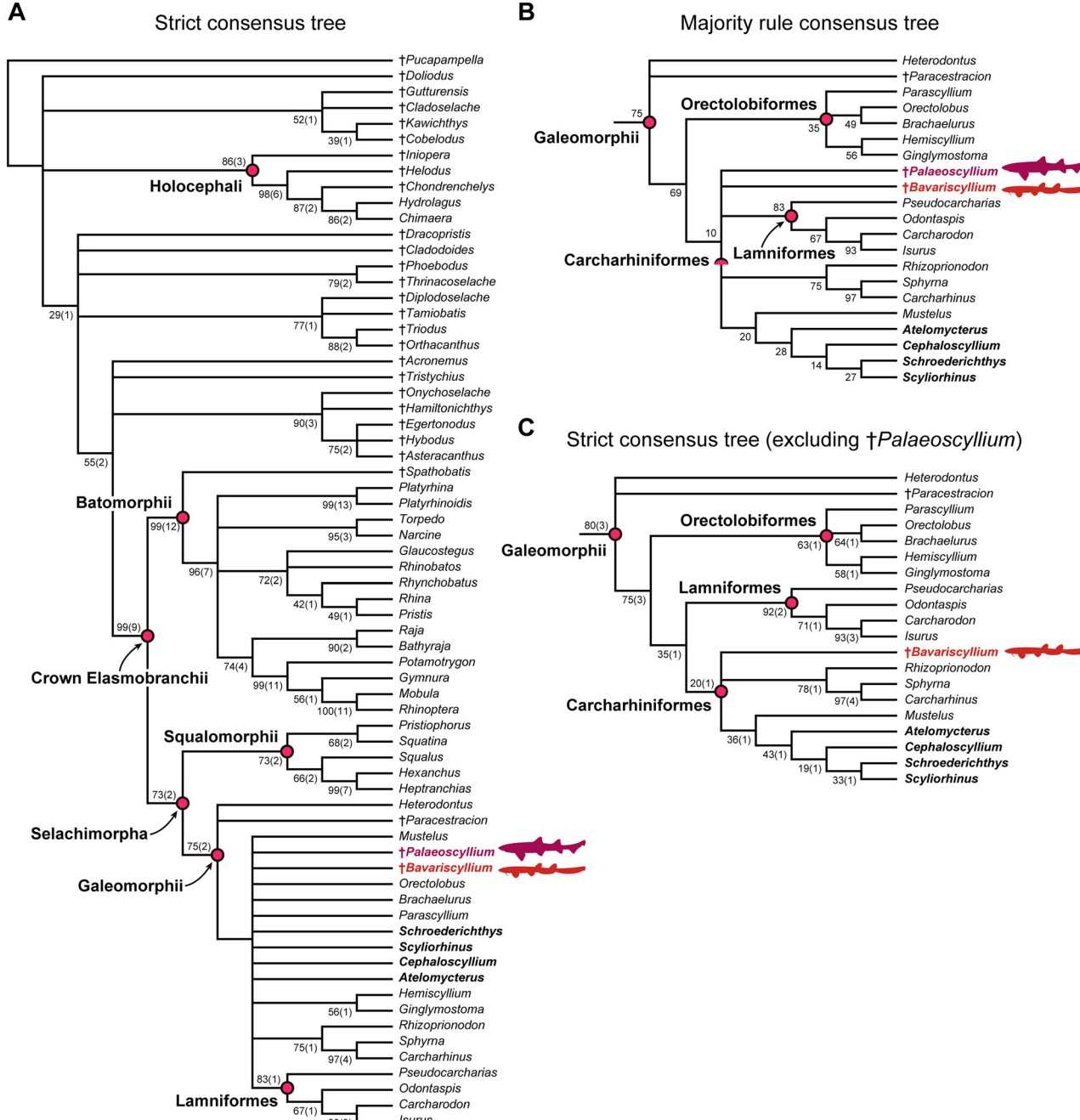

**Fig. 7 | Phylogenetic results from parsimony analysis. A** Strict consensus tree based on complete dataset; **B** reduced majority rule consensus tree with a 50% threshold showing the clade Galeomorphii; **C** reduced strict consensus tree showing the clade Galeomorphii, after removal of †*Palaeoscyllium*. Genera highlighted in bold black indicate taxa traditionally placed among catsharks in Scyliorhinidae and currently assigned either to Scyliorhinidae (*Scyliorhinus* and *Cephaloscyllium*) or Atelomycteridae (*Atelomycterus* and *Schroederichthys*), following Soares and Mathubara[32]. Bootstrap and jackknife values are shown next to nodes, with jackknife values in brackets.

characterised by a low hemiaulacorhize root exhibiting strongly flared lobes, closely resemble †*Palaeoscyllium tenuidens* from the Bathonian of England[22] and †*P. formosum* from the Kimmeridgian–Tithonian of Germany, France, and England[26,48,78]). The closest similarities are with †*Thiesus concavus* from the Valanginian of France[53], with some teeth being nearly indistinguishable, differing from †*T. concavus* only in having main cusps that are slightly more robust and less lingually bent. However, this very minor difference, along with the shared lack of labial ornamentation in some teeth, is more likely to reflect intra- rather than interspecific variation, such as gynandric and/or ontogenetic heterodonty, as documented in many extant elasmobranch species (e.g.[79–82]) and particularly well expressed in carcharhiniforms with

generalised clutching-type dentitions[16,83–85]. Consequently, due to the striking similarities between the Mahlstetten teeth and †*T. concavus*, we propose treating the latter as a junior synonym of †*B. tischlingeri*, extending its stratigraphic range into the Early Cretaceous. The two fragmentary teeth from the Kimmeridgian of northern Germany described by Thies[27], originally referred to †*Bavariscyllium* sp. and both displaying the characteristic oblique labial ridges, are likewise referred here to †*B. tischlingeri*. Additionally, †*Bavariscyllium* may also extend down into the Middle Jurassic, as suggested by a lateral tooth from the Bathonian of England[22], initially reported by Underwood and Ward[22] as Scyliorhinidae? gen. indet. and later considered by Guinot et al.[53] as potentially representing an unnamed species of

†*Thiesus*, though this cannot be confirmed until more dental material is found.

Given their shared scyliorhinid-like clutching-type dentition, it is tempting to suggest a close phylogenetic affinity between †*Bavariscyllium* and †*Palaeoscyllium*. However, their markedly different body shapes and uncertain phylogenetic placement also leave room for more distant evolutionary relationships. The shared dental similarities could therefore either be the result of convergence driven by similar feeding ecologies or represent the plesiomorphic state inherited from the last common ancestor of [[Orectolobiformes] + [Carcharhiniformes + Lamniformes]]. As a result, we recommend treating both †*Bavariscyllium* and †*Palaeoscyllium* as Galeomorphii *incerti ordinis* until more complete fossil material becomes available for further study. This phylogenetic ambiguity also extends to the purported carcharhiniforms †*Eypea leesi* and †*Praeproscyllium oxoniensis* from the Bathonian of England[22], as their tooth crowns exhibit only minor differences to those of †*Bavariscyllium* and †*Palaeoscyllium*. Notably, †*Eypea leesi* recently had been used to time-calibrate the carcharhiniform crown node in phylogenomic analyses[9,24]. However, since none of the Bathonian species can be confidently assigned to either carcharhiniform or orectolobiform sharks, their suitability as calibration fossils for dating the divergence of carcharhiniform sharks should be considered tentative until their phylogenetic relationships are resolved. This is further complicated by a single scyliorhinid-like tooth of uncertain systematic affinities from the middle Hettangian of Belgium[86,87], which suggests that †*Praeproscyllium* may extend back into the earliest Jurassic, predating the oldest unambiguous orectolobiform fossil records from the late Early Jurassic[88–90]. However, to be able to confirm this, more complete fossil material is needed.

Another galeomorph shark that may rank among the earliest carcharhiniforms is †*Corysodon*, represented by †*C. cirinensis* from the Kimmeridgian of continental Europe[25,67,91] and †*C. multicristatus* from the early Aptian of England[92]. Initially described based on two poorly preserved skeletons from the Kimmeridgian of France, †*Corysodon* is reminiscent of †*Palaeoscyllium*, but is characterised by unique fan-shaped teeth. Originally referred to orectolobiforms by Saint-Seine[67] and subsequently synonymised with †*Palaeoscyllium* by Cappetta[93], †*Corysodon* was resurrected from synonymy by Thies and Candoni[25], who tentatively considered it as Carcharhiniformes *incertae familiae*. While the teeth of †*C. cirinensis* lack both a uvula and apron and possess a hemiaulacorhize root, those of †*C. multicristatus* are distinguished by their holaulacorhize root and the presence of a characteristic boss-like labial apron. As argued by Batchelor and Duffin[92], these features do not necessarily support a closer phylogenetic relationship with orectolobiforms rather than carcharhiniforms, but neither do they contradict it.

Establishing the minimum age for the emergence of carcharhiniform sharks remains elusive due to the scarcity of suitable fossil material that could be used to trace the character evolution of early carcharhiniforms beyond dental traits. Nevertheless, a mid-Jurassic age remains feasible, given that their sister group, the lamniforms, are considered to date back to the Bathonian[94,95]. However, this hypothesis remains largely untested[20], stressing the need for further research.

The study of living sharks has shown that lifestyle can be predicted from body shape[33,96,97]. †*Bavariscyllium*, with its slender, elongate body and low caudal fin, likely adopted a bottom-dwelling lifestyle, employing an anguilliform mode of undulatory swimming[98,99] to navigate the seafloor. In contrast, †*Palaeoscyllium* and †*Corysodon*, with their slightly more robust bodies and truly heterocercal caudal fins, resemble extant littoral sharks[10,33], suggesting a carangiform swimming mode[98,99] and thus a more active and mobile lifestyle, although they may also have been capable of resting on the seafloor. Despite their morphological similarities, †*Palaeoscyllium* and †*Corysodon* likely exploited different food resources. The unique fan-shaped teeth of †*Corysodon* suggest that it was well-adapted to rasping hard-shelled prey[92], whereas the generalised scyliorhinid-like clutching-type dentition of †*Palaeoscyllium* suggests a broader dietary spectrum, as is also the case for †*Bavariscyllium*. The function of the whisker-like throat barbel in †*Bavariscyllium* remains unknown. However, by analogy with the extant parascylliid *Cirrhoscyllium japonicum*, whose cartilage-cored throat barbels lack muscle associations and sensory receptors but are innervated by sensory branches of the facial (VII) cranial nerve[41], it may have served a mechanosensory function, potentially responding to physical stimuli. Studies on extant shark body shapes[33,96,97], combined with time-calibrated phylogenomic analyses[9,24,100], suggest that most major shark body plans did not diverge before the Cretaceous. We present quantitative evidence indicating that Jurassic fusiform demersal galeomorphs, despite probably not being placed among extant orders, had already evolved a variety of body plans resembling those of their living ecological counterparts, possibly as an adaptive response to an increasing risk of niche overlap resulting from their rapid diversification[1–3]. Outside the taxa included in this dataset, more robust, less streamlined early galeomorphs, such as †*Palaeocarcharias*[101,102] and †*Paracestracion*[77], further expand this range of body forms, encompassing many of the ecological roles that demersal galeomorph sharks occupy today. This is consistent with the morphologically diverse dental fossil record, which might indicate a variety of dietary preferences among early galeomorphs[15]. Although pelagic galeomorphs would not emerge until the Cretaceous[8,103], they were exploring a wide range of demersal body plans even early in their radiation.

## Methods
### Fossil material
The articulated specimens presented in the main text come from the Late Jurassic Solnhofen Archipelago in southern Germany, which is considered a classic example of a Konservat-Lagerstätte due to its exceptional preservation[104], and refers to a number of late Kimmeridgian–early Tithonian sites located halfway between the cities of Nuremberg in the North and Munich in the South (see[20]). These localities afford access to thick packages of finely laminated limestones that are famous for having yielded an extraordinarily well-preserved plethora of marine and terrestrial organisms[104,105]. The deposits were formed in closely associated depocenters bounded by sponge and occasional coral reefs, some intermittently exposed as small islands. Restricted water circulation and limited water exchange resulted in dysoxic to anoxic bottom conditions that suppressed bioturbation, microbial decay, and scavenging. Combined with rapid burial by abiogenic carbonate precipitation, storm events, and synsedimentary mass flows, these conditions likely facilitated the exceptional fossil preservation[105].

The Solnhofen Archipelago represents one of the most productive sources of Mesozoic chondrichthyans, having yielded numerous whole-bodied specimens of sharks, rays and holocephalans (for a review, see[20]). †*Bavariscyllium tischlingeri* has thus far only been recorded from deposits referred to the early Tithonian Altmühltal Formation (previously known as the 'Solnhofener Plattenkalke'). Currently, seven articulated specimens are known, two of which are held in private collections and are not available for study[35,45,46]. Virtually all specimens are complete, except for SNSB-BSPG 1878 IV 6, which is poorly preserved and has previously been referred to as †*Synechodus* sp.[48,50,51,106]. Some specimens were examined under ultraviolet (UV) light following the methodology of Tischlinger and Arratia[107] to improve the identification of specific morphological characters.

Previously undocumented teeth of the holotype specimen of †*Bavariscyllium tischlingeri* (JME SOS 4124) shown in Fig. 2C, D, are coated with an unknown transparent consolidant, likely applied to seal and stabilise the fossil, and were photographed under 365 nm UV light using a Canon EOS 800D DSLR camera mounted with a LIUZHENZHEN NDPL-2 microscope objective on a Novex Trino Zoom. The resulting images were post-processed in Adobe Photoshop CC 2021 for improved clarity, without altering morphological details.

In addition, 17 isolated teeth have been retrieved from early Kimmeridgian marls near Mahlstetten in southern Germany, about 10 km southwest of the Nusplingen locality[20,48]. Discovered in 1995, the Mahlstetten locality has yielded a diverse chondrichthyan assemblage, comprising hundreds of isolated teeth whose taxonomic affinities have yet to be fully established[48,108].

Descriptive skeletal and dental terminology employed in this study largely follows that of Compagno[109] and Cappetta[15], respectively. All the fossil material presented in this study is held in publicly accessible collections, listed below.

## Institutional abbreviations

JME, Jura-Museum Eichstätt, Eichstätt, Germany; LF, Lauer Foundation for Paleontology, Science and Education, Wheaton, Illinois, USA; SMNK, Staatliches Museum für Naturkunde Karlsruhe, Germany; SMF, Senckenberg Naturmuseum, Frankfurt, Germany; SMNS, Staatliches Museum für Naturkunde Stuttgart, Germany; SNSB-BSPG, Bayerische Staatssammlung für Paläontologie und Geologie, Munich, Germany.

## Morphometric analysis

Using 16 body measurements (see Supplementary Fig. 1), we compared four holomorphic (i.e. virtually complete articulated) specimens of †*Bavariscyllium tischlingeri* (JME SOS 4124, LF 1436, SMF P 272, SMNS 96086) and two holomorphic specimens of †*Palaeoscyllium formosum* (SMNK-PAL 44950, SNSB-BSPG AS I 589, Supplementary Fig. 2) with 160 extant carcharhiniform and 25 orectolobiform species (Supplementary Data 1). These covered five families of phenotypically similar sharks, including catsharks, divided into Scyliorhinidae, Atelomycteridae and Pentanchidae (following Soares and Mathubara[32]), plus Parascylliidae and Hemiscylliidae. Not included in the analysis were †*Corysodon cirinensis* and †*Pararhincodon lehmani* due to lack of suitable fossil material. Measurements, expressed as percentages of the total length (TL) of each specimen, were taken using ImageJ v1.53t, with each measurement repeated three times and averaged to minimise potential errors. Building upon previous studies (e.g.[8,103,110,111]), morphometric measurements of extant shark species were obtained through illustrations from Ebert's et al.[4] *Sharks of the World: A Complete Guide*. Missing values (one specimen of †*B. tischlingeri* [SMNS 96086] did not allow all measurements to be taken due to its dorsoventral preservation) were imputed using a regularised iterative principal component analysis (PCA) algorithm with the function imputePCA in the R package *missMDA*[112]. The resulting dataset was subjected to a PCA in order to determine major axes of variation.

A series of statistical tests was conducted to evaluate intergroup differences. Initially, a Shapiro-Wilk test for normality was applied to each measurement. Subsequently, subsets containing either normally or non-normally distributed measurements were created. The subset containing non-normally distributed measurements underwent a Kruskal–Wallis test, followed by a pairwise Wilcoxon rank-sum test with Bonferroni corrections. Normally distributed measurements were subjected to an analysis of variance (ANOVA) and then to pairwise comparisons.

In addition to the PCA, a linear discriminant analysis (LDA) was conducted to explore the a priori classification using the leave-one-out cross-validation (LOOCV) approach. In total, eleven dimensions were retained and subjected to a LDA using the LDA function in the R package *Momocs*[113]. Subsequently, the variables generated by the PCA were employed to estimate morphological disparity as a sum of variances utilising the R package *dispRity*[114]. To account for differences in sample size, we applied 100 bootstrap iterations using the dispRity.per.group function. In addition, disparity was also calculated as the sum of ranges, which is more sensitive to sample size variation. Finally, a pairwise Wilcoxon rank-sum test with Bonferroni corrections was employed to test for differences in disparity.

All plots presented in the text were created using the R packages *ggplot2*[115], *ggpubr*[116], and *viridisLite*[117] and finalised using Adobe Illustrator CC 2021.

## Phylogenetic analysis

To test the phylogenetic relationships of both †*Bavariscyllium* and †*Palaeoscyllium*, we incorporated them into a slightly amended and modified cladistic character-taxon matrix of Vullo et al.[8]. This resulted in a dataset containing 212 morphological characters scored for 68 operational taxonomic units (OTUs) (Supplementary Data 2). The character-taxon matrix was compiled in Mesquite 3.81[118] and analysed using maximum parsimony in TNT v.1.6[119] under Goloboff et al.'s[120] protocol for characters with logical dependencies.

The search for the most parsimonious trees (MPTs) was conducted with the deactivation of invariable characters under the collapse rule 3 (default for TNT). The search parameters included a combination of 1000 ratchet iterations and five rounds of tree fusion, with three replications. The search was performed ten times with different random seed numbers to ensure an adequate sampling of the tree space. Given that the searches yielded numerous MPTs but produced the same consensus tree, an additional search was conducted using the same parameters, this time with the command "bbreak" applied to estimate additional trees of the same length. All recovered trees were retained and used to produce a strict consensus tree. Due to the large amount of missing data in †*Palaeoscyllium*, which led to the collapse of several galeomorph clades, the above search protocol was run a second time, with †*Palaeoscyllium* excluded. Jackknife and Bremer analyses were employed to estimate clade support.

## Reporting summary

Further information on research design is available in the Nature Portfolio Reporting Summary linked to this article.

## Data availability

All data supporting the findings of this study are provided in the paper and its Supplementary Information. The fossil material presented in this study is housed in publicly accessible collections, with details available in the paper and its Supplementary Information.

## Code availability

R code for morphometrics and TNT character-taxon matrix are publicly available for download from the online repository Figshare at https://doi.org/10.6084/m9.figshare.30465527.

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

## Acknowledgements

We are grateful to Christina Ifrim (JME), Erin Maxwell (SMNS), Oliver Rauhut (SNSB-BSPG) and Julien Kimmig (SMNK) for granting access to the collections in their care. Special thanks to Rainer Brocke and Sven Tränkner (both SMF) for providing photographs of specimen SMF P 272. We also thank Alan Pradel (Museum national d'Histoire naturelle, Paris, France) for his assistance in obtaining photographs of †*Pararhincodon lehmani*, and Stefanie Klug (University of Göttingen, Germany) for the SEM images. Our deepest appreciation goes to Frederik Spindler (PALAEONAVIX, Kipfenberg, Germany) for his artistic life reconstruction of †*Bavariscyllium*. Additional support was provided by Udo Resch, Helmut Tischlinger, Charlie Underwood and Patrick Jambura. The constructive feedback from three anonymous reviewers improved the quality of the manuscript. This research was funded in part by the Austrian Science Fund (FWF, P 35357; grant https://doi.org/10.55776/P35357) to J.K. F.A.L.-R. is supported by a postdoctoral fellowship from DGAPA-UNAM (No. CJIC/CTIC/5475/2023). R.P.D. is supported by funding from the European Union's Horizon Europe research and innovation programme under a Marie Skłodowska-Curie grant agreement (No. 101062426). For open access purposes, the authors have applied a CC BY public copyright license to any author accepted manuscript version arising from this submission.

## Author contributions

S.S.: conceptualization, methodology, formal analysis, investigation, visualization, writing—original draft, writing—review and editing; J.T.: conceptualization, methodology, formal analysis, investigation, writing—review and editing; F.A.L.-R.: methodology, formal analysis, investigation; writing—review and editing; E.V.-S.: conceptualization, methodology, formal analysis, investigation, writing—review and editing; A.B.: investigation, writing—review and editing; M.A.: investigation, writing—review and editing; R.P.D.: investigation, writing—review and editing; B.L.: investigation, writing—review and editing; R.L.: investigation, writing—review and editing; A.H.: investigation, writing—review and editing; J.K.: conceptualization, investigation, resources, writing—review and editing.

## Competing interests

The authors declare no competing interests.
