## [Transparent Peer Review file · Communications Biology]

Reappraisal of the extinct barbelthroat shark †*Bavariscyllium* and the nebulous origin of carcharhiniform galeomorphs

Corresponding Author: Professor Jürgen Kriwet

Version 0:

Reviewer comments:

Reviewer #1

(Remarks to the Author)

This well-written manuscript presents a study of fossil shark specimens that are exceptional for their quality of preservation and age. These specimens are key fossils for understanding the evolution of modern elasmobranchs and their initial radiation during the Jurassic period. While the results may not be totally novel they will be of interest to the paleoichthyological community for applying analytical methods. This manuscript is part of the ongoing work of this team, which aims to re-examine and revise the systematics of old specimens collected in Late Jurassic Solnhofen Archipelago in southern Germany (e.g. Kriwet & Klug, 2005; Lopez-Romero et al., 2020; Stumph et al., 2022; Jumbura et al., 2023), whether from public or private collections, which are now available to researchers. In this respect, the aim of this re-study of elasmobranch skeletons of enigmatic species *Bavariscyllium tischlingeri* using two analytical approaches (morphometry, phylogeny) is really exciting.

However, the manuscript contains flaws and/or shortcuts that make the results questionable and the systematic review, in my opinion, unsatisfactory. The main criticisms are essentially due to flawed nested reasoning, namely:

- that (1), the authors partially revised diagnosis and systematic affiliation of *Bavariscyllium tischlingeri* from the Solnhofen region by first attributing teeth from another region and another age without any serious argument or study. The proposed new diagnosis of this species is based in particular on these dental characteristics observed on isolated teeth from another locality ("comprising hundreds of isolated teeth whose taxonomic affinities have yet to be fully established by Kriwet & Klug 2004; Klug 2009") collected in deposit of older age. The authors justify this systematic comparison in a few words by a global resemblance (unproven in my opinion) to the rare teeth of the holotype, as published by Thies & Leidner in 2011.

- That (2), the authors have excluded the possibility that these 4 semi-complete specimens (<25 cm) are juveniles, which would invalidate in part their comparative morphometric analysis (not verifiable due to missing data in supplementary data where measurements are hidden with #####) and their phylogenetic analysis.

- That (3), The authors do not really describe the skeletal anatomy of the specimens in any way. This makes it difficult to understand the coding of characters (phylogeny) and the main result (placement close to the Carcharhiniformes). We can only verify this by looking at the teeth, but unfortunately these do not belong to the holotype or the associated specimens (see point 1). I haven't been able to work out how the characters were coded from the figures as available in the MS.

I have listed below some questions/accuracies and the main points/arguments that I found problematic and need to be checked by the authors.

Anatomy

Barbels: why a single? why there is no pair of barbels as expected in sharks. Preservation artefacts resulting from lateral compression could erase the second one, note it is on the right in the three right flanked specimens and on the left in the left flanked specimen (and missing in the dorsoventral specimen)? Regarding the right/left position of the barbels, depending on the side of observation, we can assume that there are two, as for the pectoral and pelvic fins?

Gender: Authors argued that all specimens are (?adult) female, lacking visible claspers on specimens. Why not all juveniles? as all specimens are small (<25cm TL) and that all specimens come from the lagoon area of Solnhofen. This

point, which is completely overlooked, is crucial for the rest of the analysis. In fact, the analysis of juveniles becomes complicated, both morphometrically and phylogenetically, because it encounters problems of allometry that affect both biometric and phylogenetic criteria, especially since the living specimens used for comparison are all adults.

Teeth: Attribution of several isolated teeth from another locality (Mahlstetten) and another age (here, Kimmeridgian) to this Tithonian taxon from Eichstätt (Solnhofen area) is extremely problematic for me, especially because they are used to modify original diagnosis, to code morphological characters and support phylogenetic analysis. This cannot be done because there is nothing to indicate with certainty that these teeth belong to this species (and also to this genus).

The authors avoid to discuss about this, considering in one short sentence that they resemble the holotype illustrated by Thies & Leidner (2011). However, they appear to be quite different from teeth belonging to holotype of *Bavariscyllium tischlingeri* Thies, 2005 (SOS 4124), from the Eichstätt Tithonian and well-illustrated by Thies & Leidner (2011: plate 45). The teeth figured by Thies & Leidner (2011) are of lesser quality, the roots of the teeth still being absent, but they have the merit of belonging to the holotype (extracted from jaws). The dentition of the holotype is much less heterodont than that shown by the authors from the dentition reconstituted from isolated teeth coming from Kimmeridgian of Mahlstetten deposit, the posterior teeth being less spread out. The cusps of the anterior and anterolateral teeth of holotype are much more slender, high and developed than those of the Kimmeridgian, and the heels never show well-formed denticles, whereas those of the Kimmeridgian show up to two pairs of well-detached and sometimes divergent secondary denticles. The main cusp is much more curved lingually in Tithonian teeth, whereas the Kimmeridgian cusps have a fairly flat labial surface. The ornamentation of the crown is much weaker on the teeth of the holotype compared to those of the Kimmeridgian. The roots are not known on the teeth of the holotype, so it is difficult to compare them with those of the Kimmeridgian. Even if the authors do not agree with my previous remarks, they cannot attribute isolated Kimmeridgian teeth to a Tithonian species without a more solid argument and, above all, use them to redefine the species. To conclude, we cannot yet be sure that the Kimmeridgian teeth belong to this species. The Lower Kimmeridgian teeth from Mahlstetten are much more similar to species *Thiesus concavus* from Cretaceous than they are to *Bavariscyllium tischlingeri*, as Thies & Leidner (2011) figure. The authors discuss this last great resemblance, but later in the manuscript, by placing the cretaceous species *Thiesus concavus* as a synonym of Jurassic *Bavariscyllium tischlingeri*, omitting that the Kimmeridgian teeth could also belong to *Thiesus* and not to *Bavariscyllium*...

Actually, all the dental features used in the updated diagnosis and phylogenetical analysis (characters 199-212) need to be removed to be totally accurate and serious. Only the teeth described by Thies & Leidner (2011) can be safely said to belong to species, and only these can be used to identify, describe and used for coding *Bavariscyllium tischlingeri*.

Morphometry

As mentioned above, it is quite possible that the four Solnhofen fossils are juveniles, because they are very small (less than 25 cm) and the way the area used to be (the palaeoenvironment) suggests that it was a lagoon, which is a great place for young animals to grow up. Unfortunately, the measurements of the fossils (and those scanned from Book of Ebert, 2021) are not available in the supplementary data. I cannot check for a possible effect of size and/or allometry. Whatever, this possibility to have only juveniles is never discussed in MS.

I also wondered why *Bavariscyllium tischlingeri* is compared only with the coeval species *Paleoscyllium formosum*, and not with others such as the coeval orectolobiform "*Crossorhinus jurassicus*", the dubious "*Palaeoscyllium minus*" from Eichstätt (see Thies & Leidner, 2011), or the possible orectolobiform *Paleocarcharias stomeri* from Blumenberg also studied by the same team? It would make sense to use both a more probable carcharhiniform like *Paleoscyllium formosum* and a potential orectolobiform from these same deposits/ages to compare *Bavariscyllium* from elements that have undergone comparable deformation/compression.

Even unnecessary with PCA based on covariance, are the different sizes normalised by TL? If not normalised by TL, the large variance induced by a large heterogeneity in comparative sample could minimise some effects on small specimens such as *Bavariscyllium tischlingeri*.

The authors suggested ontogenetic allometry for *Palaeoscyllium formosum* because the large separation between the two *Palaeoscyllium* specimens is intriguing and could be due to ontogenetic allometry, as has been documented in extant sharks (e.g. Ahnelt et al. 2020; Sternes & Higham 2022; Gayford et al. 2024). This is certainly possible, and it would have been interesting if the authors had asked the same questions about the target species *Bavariscyllium tischlingeri*.

The authors successfully tested which measurements were normally and non-normally distributed using a variety of statistical approaches. However, it is not clear why they do not use only these last characters to study the position of their two fossil taxa. We can expect that measurements normally distributed probably covariate as the whole. The authors also used morphological disparity (Figure 5), Can they specify what they are analysing? covariance matrix divided by the number of observations in the group?

Phylogeny

I am a bit surprised that authors do not have a better resolution with such large matrix, whereas Vullo et al., 2024 have a well resolved strict consensus for the Galeomorphi with 6349 trees? For instance, why you remove some fossils from the same deposits (e.g. stem *Squalomorphi* Protospinax) and not others (stem *Galeomorphi* Paracestracion)? Is there a reason for this omission? The matrix of Vullo et al. 2024 is in turn taken from Jamboura et al. 2023, which in turn is taken from Landemaine et al. 2018 and Villalobos-Segura et al. 2022 etc,... with some new modifications by Authors that do not

specifically concern the requirements for the study of target species (and coded as ?)...? Why? On the other hand, the presence/absence of barbels, a character much emphasised by authors on fossil material, is not even used in the matrix, Why ???? It could possibly give an advantage for a placement with the Orectolobiformes that possess such feature. Regarding the synapomorphic characters of the clade (Carcharhiniformes and Bavariscyllium), I have noticed that very little has been documented on fossil taxa, suggesting that the groupings are essentially based on homoplasy : - e.g. characters 17[0], 33[4], 123[1], 174[1] (synapomorphy of Carcharhiniforme + Bavariscyllium), only character 123[1] is really coded (presence of a Scapular_posterior_process) for Bavariscyllium. Bavariscyllium is therefore only related to Carcharhiniformes on the basis of homoplasy and the presence of a posterodorsal process at base of scapular process (pd). Da Silva et al. (2017) suggested that presence of two posterior triangular process of scapula (pdt and pvt) could be characteristic of "stem" Carcharhiniforms. They hypothesized that the posterior ventral (pvt) and dorsal (pdt) triangular processes found in Scyliorhinidae, Triakidae and Proscylliidae are multiple independent acquisitions within Carcharhiniformes. Have you seen these peculiar triangular processes on the four specimens? - or e.g. characters 3[2], 9[0], 114[0], 135[0], 203[0] (synapomorphy of the clade (Carchar+Lamnif+Bavariscyllium+Palaeoscyllium), so that only character 203[0] = orthodentine, is coded on the teeth supposed to belong to Bavariscyllium (see previous chapter). To conclude, the position of Bavariscyllium close to the Carcharhiniformes (also previously attributed to this Order by formal author Thies, 2005) known from Tithonian skeletons (including the holotype), is based on an ambiguous character (123[1]) or on teeth that come from another locality and age rather than on teeth from holotype.

I am aware of the difficulty of studying fossil specimens, and that the anatomical mastery of authors is much greater than my own, but it is not possible to judge these characters coded in this way in the absence of sufficiently precise illustrations of the SNSB-BSPG 1878 IV 6, which nevertheless seems to have provided the maximum data. To complete my questioning, and after deciphering the matrix in the appendices : Bavariscyllium ???0????????????????001????????01????????01?0?1?? 0?????????1?????????11??11011?????00?????1??11?2??00?????1?????0?0??10100000100---34213????001????? 01??0?11?1-11000?10010210000?????????????0---10000---0100000?0000001

I have attempted to visualise the characters thus observed and coded for Bavariscyllium tischlingeri and I can't see/understand on the images provided how certain characters could have been coded (or possibly inferred) on these specimens whose skeleton pieces are not very well preserved (unlike the body print)? Do you have any other elements and/or other images that would make it possible to see and encode these characters? such as character 4 [0] that concerns the internasal plate, 21-22 (Interorbital_space / Optic_pedicel), 30 (Occipital_crest), 31(Basicranial_processes), 41 (Lateral_otic_process), 42(Postotic_process), 44 (Stapedial_arch), 49 (Metotic_otic_occipital_fissure), 59, 68-69 (e.g. Glossopharyngeal_nerve_path)... Only obvious skeletal/body print characters (such as 189-193 : Placoid_scales / Malar_and_alar_thorns / Lateral_rostral_dermal_denticles / up_Dorsal_fin_spines / Dorsal_Cephalic_spines ; 198-212 : Serrated_tail_sting) appear to be easily codable on these specimens. In cases where fossils are too uninformative/uncodable, why not use fewer characters (less ambiguous and/or observable) but apply a molecular backbone constraint to the extant taxa in previously published phylogenies?

Reviewer #2

(Remarks to the Author)

The manuscript title "Reappraisal of the extinct barbelthroat shark †Bavariscyllium and the nebulous origin of carcharhiniform galeomorphs" is a very interesting work in which the author review 5 articulated specimens and 17 isolated teeth belonging to the genus †Bavariscyllium in order to review the diagnosis of this genus. They also conducted some morphometric and phylogenetical analyses to test if this genus belongs to Carcharhiniformes or Orectolobiformes, and to try to elucidate early galeomorph evolution. Despite their extensive study, the authors couldn't assign this genus to any of the orders mentioned above, leaving them as Incerti ordinis and Incertae familiae. They stress the need for more research in order to untangled the placement of †Bavariscyllium, †Palaeoscyllium, †Corysodon, and some other species with uncertain affinities in the phylogeny and the early evolution of galeomorph sharks. Research like this is very important, because as the authors have stated, the great variability of dental traits in sharks sometimes cannot be use to infer phylogenetical placement, as convergent evolution could be the more parsimonious explanation for observed similarities. The inclusion of morphological data, although scarce in the fossil record, is very valuable for current and future research in which more suitable fossil material can be used in combination with the dental characters.

Overall, the authors have presented a well-researched work, an provided all the necessary data, and information about the software and packages used to replicate the results they provided. The manuscript is well-written, the quality of the figures is excellent and support most of the claims the authors made in the main text. The graphs are readable and informative, and the statistical analysis are adequate and support their conclusions.

Despite all the expose above, I have some comments and suggestions that I would like to ask the authors to consider. I have listed some of the most important here, but there are some more comments in the PDF attached:

1. My main disagreement with the authors is the description of the isolated teeth. The authors state that "additional third pair of very small rudimentary lateral cusplets may be present" (lines 280-281), and also that "below the collar where the lateral root lobes join, there is a row of small foramina, and additional foramina may extend laterally along the length of the lateral heels" (lines 296-298). None of these features can be seen in the pictures provided by the authors: only two pair of lateral cusplets is depicted in all the teeth (with exception of the Fig. 3 G-H, that only shows one pair); and although most of the teeth preserve the root, I cannot distinguish the row of small foramina that the authors describe. Could the authors provide images in which this features appear, or highlight these features in the pictures already provided?

2. Line 230: I suggest to change the sentence “the splanchnocranium is visible in one specimen” to “the splanchnocranium is visible in specimen XXX”

3. Line 311: Please, specify in which figure of Thies & Leidner (2011) are figured the dermal denticles. This is a very long paper with multitude of pictures, so to facilitate the readers to locate those dermal denticles more quickly.

4. In my opinion, Figure 4A doesn't add much information in its current format, as all the information is in the supplementary material. For that reason, I suggest the authors to: a) put Supplementary Figure 1 as Figure 4A, or b) delete completely Figure 4A and refer only to Supplementary Figure 1.

5. In general, I recommend to the authors to cite more their figures in the whole description section. Please, some examples can be found in the annotated PDF attached.

6. Some of the references are missing for the main text or the references section. There are also some discrepancies in the dates for some of them. Please, see the comments in the attached PDF.

It is for all of the above comments, that I recommend some minor revisions to improve the manuscript before it can be published

Reviewer #3

(Remarks to the Author)

Key results:

The authors present a re-description of the Jurassic elasmobranch shark *Bavariscyllium*.

The revision has implications for how we understand the evolutionary radiation (morphospace occupation, time-tree node calibration) of major living groups – lamniform and orectolobiform sharks, although placement of this genus within the lamniform-carcharhiniform group is uncertain/unstable. More informatively, the body-shape and dentition indicate early ecomorphological specialization within the clade.

Originality and significance:

The paper is a significant addition to a rapidly improving dataset documenting the historical biodiversity of a major division of extant vertebrates. Associating carefully described and quantified holomorphic fossil taxa with the vast record of isolated shark teeth is a significant addition in itself -testing tooth-based diagnoses and attendant hypotheses of spatial-temporal taxon ranges. The careful treatment of phylogenetic output, coupled with morphometric analyses, are rarely seen applied to Mesozoic sharks.

Data & methodology:

No problems – standard methods employed.... Matrix and new characters listed, supplied, and justified (a refreshing change).

Conclusions:

See above.

Suggested improvements:

See the specific comments below.

Most importantly – add line drawings to explain/guide interpretation of skeletal remains in photographs.

Pay more attention (in text) to details of appendicular and axial skeleton.

Add a skeletal reconstruction, insofar as one can be completed.

See questions raised about use of data lodged in private/semi private collections.

Consider changing the title! 'The nebulous origin...' kills interest - it suggests you've found nothing very much, which is not true.

Specific comments:

L.109. Is it possible to show the former/current membership of Scyliorhinidae in Figure 6?

-number the figures for ease of identification

L.121. Barbels as sense organs – additional information on kinds of sensory receptors known to be present – chemosensory?

L.151-174 – Diagnosis for genus lengthy, but most notable for more than half of its content characterizing details of the teeth. Does this indicate a largely tooth-determined phylogenetic placement? Can the diagnosis be condensed to include the most diagnostic features of the teeth?

Figure 1 – clear, strong presentation. However, a specimen from private or semi-private (? Note comment on line 224) collection included – Lauer Foundation. The issue here concerns long term access for research and reproducibility. Is the fate of this specimen (and others) agreed after the lifetime of the present collection, i.e. will it be sold or is there an agreed donation to a fully accredited public museum collection?

Figure 2. Clear photographs, but these prompted searches for traces of restored features such as the spiracle (given the completeness of the squamation).

Line drawings of best estimates of endoskeletal cartilages (components of mandibular and hyoid arches, pectoral and pelvic skeletons; axial column) would add greatly to the value of this manuscript. Low angle illumination to complement the UV illuminated details would be a great help.

Ideally, it would be good to see skeletal reconstruction, as far as it can be determined.

L. 237 – Paired scapulocoracoids fused – this is the kind of detail that deserves support from a line-drawn interpretation of Fig. 2D (for example).

L. 240+ - Proximal radials (pro, meso, & metapterygium) of the pectoral fin – again, an appeal for line-drawing of the specimen. Adding thick black lines to the photograph is not a good idea (reminiscent of fossils where the boundaries of bones have been painted-in by well-intentioned 19th and early 20th century researchers).

Distal radials? No estimate of minimum number and distribution, even if thought to be incomplete.

L. 244+ Pelvic skeleton – similar issues. How are the fluorescing patches in the photographs interpreted? For example ‘the basipterygium appears to have been straight’ – without a diagram I’m unsure about the basis for this interpretation.

L. 252+. And so on for the dorsal fin skeletons.

L. 262. Extremely brief treatment of the axial skeleton. Centra mentioned – but no counts; estimates of numbers per body region, or related to, say, levels of leading edges of fins?

L. 265. The teeth from Mahlstetten – excellent images, but for the reader new to these sharks, it would be helpful if this plate included an example from the Solnhofen holomorphs – anchoring tooth-morphs to body. What uniquely derived characteristics, or unique combination of general features, are shared?

For tooth terminology, standard, cite Capetta '87 – or some other source appropriate to task.

L. 299+ How are the monognathal heterodontal trends known, if complete teeth are only found isolated, from a different locality. Is there a jaw with a more-or-less complete generation of emerged teeth, available for inspection. Is it possible that heterodonty of the kind depicted in Fig. 3 might occur within a single generative tooth set (one family)?

L. 306. ‘The dermal denticle’. No, there’s more than one, and likely more than one kind of crown. Have Thies & Leidner (2011) document dermal denticle variation, sampling from the rostrum and/or leading edges of fins, and compared denticles from dorsal and ventral surfaces of the trunk?

L. 314 To assess affinity? ‘Affinity’ suggests body shapes are being compared to establish galeomorph interrelationships. Surely the question here is to discover whether established clades occupy distinct areas of morphospace and how these align with hypotheses of Bavariscyllium affinity. However, the questions (one or more) behind this morphometrics section are not clear. The current text reads as if the morphometrics were done because it’s the kind of thing that these sorts of papers include... and then some ideas/observations were written about the graph plots. Comments on the positions of dorsal fins (lines 345-347) don’t require morphometric analyses.

L. 367. What would it mean here to be ‘allied’ with any family? Phenotypic resemblance – signifying what?

Phylogenetic relationships

I’ve not had time to run the analysis – but pleased to see all relevant materials available in supplementary data.

L. 380 ‘evolutionary steps’ – these steps are estimates of character state transitions.

For the majority rule consensus tree, what % threshold? (perhaps mentioned in supplementary data – but simple to insert in the figure legend).

Fig. 6 – supplementary data states that jackknife and Bremer support values were calculated.

These could be usefully added to Fig. 6A.

Discussion

L. 425 – Not all orectolobiforms have the interbasal space– see Cephaloscyllium -I thought (perhaps mistakenly) that this was the plesiomorphic condition for the group. What might these different kinds of fin skeleton tell us about fin function?

L. 520+ Final paragraph on body shapes and Jurassic diversification broadens the discussion – for now, this is where the interest value might lie.

Perhaps re-order the discussion – move this section before the discussion of tooth shapes and possible diets.

And there’s always the perennial debate – which comes first – novel feeding apparatus or body shape?

Version 1:

Reviewer comments:

(Remarks to the Author)

The authors have made efforts to address some of the issues highlighted in the initial review. My primary concerns were as follows: the unequivocal attribution of material from a different deposit and age to this species, which is known from skeletons and associated teeth as originally described by Thies (2005); the possibility that the four original skeletons, each measuring less than 25 cm, were more likely juveniles rather than adults, necessitating consideration of potential allometric effects in morphometric analyses; and the challenge of interpreting the coding of morphological characters on specimens, which underpins phylogenetic analyses. While the latter issue has been ameliorated in the revised manuscript, with the inclusion of interesting appendices, the other two concerns remain inadequately addressed. The authors have attempted to justify their first choices without presenting any novel or conclusive arguments.

Additionally, other minor points have not been fully considered, such as the highly unusual presence of a single (throat) barbel in this fossil species, despite evidence suggesting the existence of two. If the authors choose to maintain their first assertion that this fossil shark possesses a single throat barbel — a characteristic not previously observed in any living and fossil species—they should either: consider the possibility that *Bavariscyllium* lacks also paired pelvic fins, as these have not been observed concurrently on the four specimens; or, alternatively, that they must exclude this strange feature (a unique whisker-like barbel as originating from the ventral surface of the throat) from the “amended diagnosis” if its presence is considered as uncertain, as it is “awaiting substantiation by additional fossil evidence (newly noted).

Concerning the second major disagreement (state of development of skeletons) I don't understand the authors' statement: “Despite the small size of the available specimens and the absence of claspers in any of them, which might suggest juvenile stages, they are interpreted as representing subadult to adult individuals. This interpretation is based on the apparently advanced mineralization of both their endo- and exoskeletons, in conjunction with the prolonged skeletogenesis observed in extant sharks, which starts during late embryonic development and continues well beyond hatching,” is unclear to me. The study by López-Romero et al. (2022) effectively illustrates ontogenetic variations in mandible morphology between a carcharhiniform and an orectolobiform. Nonetheless, this research underscores the importance of timing in developmental processes among elasmobranchs and highlights the divergence in shape that occurs even before cartilage differentiation, which is distinct from the issues addressed in the current discussion. It is unequivocal that these specimens are not neonates hatched from eggs; however, as the authors note, there exists a substantial developmental gap between the neonate and adult stages. Their concept of “advanced endo- and exoskeleton mineralization” to justify the adult stage remains unclear to me. López-Romero et al. (2022) only suggested that presence of labial cartilage could be used as evidence of advanced development stage toward adult, I am not sure that these labial cartilage are available or visible in the four skeletons? Claspers, for instance, only manifest in males upon reaching sexual maturity. It is plausible that the specimens in question are juveniles that have not yet attained sexual maturity. With a supposed adult length of 19-25 cm, *Bavariscyllium* would represent the smallest known species within the Carcharhiniformes/Orectolobiformes clade (currently, *Apristurus* and *Cirrhoscyllium* measure approximately max 30 cm in length), which is significant as these are the earliest representatives of the clade. It is acknowledged that, in terms of overall body dimensions, juveniles are approximately geometrically similar to adults (e.g., Irschick & Hammerschlag 2014). Juveniles are typically characterized by having relatively larger caudal fins than adults, a fact supported by numerous studies (see also Irschick & Hammerschlag 2014), which could potentially affect morphometric outcomes. Upon reviewing the newly available morphometric data, it appears that the four *Bavariscyllium* specimens possess the largest caudal fins (LLL with 28-29% TL or CVM with 31-33% TL) among the 185 living species analyzed. It is important to consider the juvenile state as a potential source of bias in morphometric analysis.

Concerning the first major disagreement (attribution of Kimmeridgian teeth to these Tithonian specimens) the authors do not address the question of whether these teeth can be definitively attributed to this species. I remain unconvinced of such attribution as they remain also convinced of this proximity: “While we were already confident in our conclusion prior to obtaining these images, the new findings now provide definitive support for our assertion that the Mahlstetten teeth can be unambiguously referred to †*Bavariscyllium tischlingerii*”. Despite efforts to newly figure teeth of the analysed skeletons, the question of whether the Kimmeridgian teeth are identical to the holotype described by Thies (2005) remains unresolved for me. Although the overall morphology of the new figuration (Figure 2C-D) of the “two” teeth is partially identifiable for meaningful morphological assessment, certain features are clearly absent. For instance, neither two lateral denticles nor pronounced ornamentation of the lingual face (as visible in most of the Kimmeridgian teeth) are observed in Thies' material (2005) or in the new Figure 2C-D. The inaccessibility of the holotype teeth's morphology due to the consolidant is indeed unfortunate, as it could have facilitated confirmation or refinement of the match. Regrettably, even with this new representation, it remains challenging to classify the Kimmeridgian material within this species, and even more so to be used as support of the new ‘amended diagnosis’ of this taxon, which is formally based on a Tithonian holotype with already figured teeth.

Currently, the Kimmeridgian teeth could also be attributable to †*Palaeoscyllium tenuidens* from the Bathonian of England (Underwood & Ward 2004), to †*P. formosum* from the Kimmeridgian-Tithonian of Germany, France, and England (Candoni 1993; Underwood 2002; Kriwet & Klug 2004), or even to †*Thiesus concavus* from the Valanginian of France (Guinot et al. 2014), as they further elaborate. The authors propose a convoluted process of synonymy, wherein indeterminate isolated teeth are assigned to a known species, subsequently leading to the synonymization of other, more recent species based on isolated teeth. To substantiate my uncertainty, I also reviewed that Thies (2005) has further identified Kimmeridgian teeth from another deposit in Germany, attributing them to the genus *Bavariscyllium*, though he refrains from assigning them to the sole species within the genus. These teeth exhibit a closer resemblance to the holotype than to those illustrated by the authors (not discussed in this manuscript, but included in the species reference list). I have included a figure compiled for

this purpose, featuring the teeth of the *Bavariscyllium* holotype, those from Kimmeridgian and newly attributed to the same species, those assigned to the *Bavariscyllium* (sp.) by Thies (2005) from Kimmeridgian also, as well as those from other contemporary and/or synonymized genera as identified by the authors. At present, the task of associating nearly all of these teeth (excepted that of *Palaeoscyllium*) with a single species *Bavariscyllium tischlingeri* presents a challenge that remains somewhat elusive.

If the authors wish to code their reconstructed dentition from Kimmeridgian of Mahlstetten as a character of *Bavariscyllium* for phylogeny, they may do so as material related to this genus and as an element of interest, albeit with a reservation regarding their systematic attribution (similarly with *Thiesus concavus*, which they consider a synonym), which would not fundamentally alter the results of the phylogenetic and morphometric study. However, it would be more appropriate not to include the teeth in the diagnosis, given the considerable uncertainty regarding their taxonomic affiliation. The diversity of Jurassic elasmobranchs, particularly *Carcharhiniformes*, is limited and complex, as the material is fragmentary and morphologically not very diverse (mainly belonging to scyliorhinid-like forms). It would be regrettable to consolidate them all around a few known taxa. Reviewer 3 suggested including the teeth of the holotype. As the authors rightly note, these teeth are well known and have been illustrated in detail in previous works by Thies (2005), Thies & Leidner (2011), and Cappetta (2012). However, I believe it would be beneficial for the reader to compare them with the Kimmeridgian teeth attributed to this taxon by authors.

I also have other disagreements with the authors' response regarding the use of backbones in phylogeny (e.g. "in phylogenetic studies, the use of topological constraints may obscure phylogenetic uncertainty within the dataset."). This issue is particularly pertinent when integrating modern taxa with an ancient fossil ant, as employing topological constraints is arguably the only effective way to merge robust neontological (molecular) data with the current phylogenetic framework. Although it is recognized that this method may polarize certain morphological character states a posteriori, it remains a persuasive approach. However, since the phylogenetic resolution of fossil taxa, as presented by the authors, remains unclear, I see no reason to oppose preserving this ambiguity.

Reviewer #2

(Remarks to the Author)

The authors have done an excellent work addressing all my comments and suggestions in this new version of the manuscript titled "Reappraisal of the extinct barbelthroat shark †*Bavariscyllium* and the nebulous origin of carchariform galeomorphs". The new figure for the teeth show all the features the authors describe; the figures are cited more frequently in the main text and the references has been revised.

Overall, the quality of the manuscript has improved significantly. The descriptions of the external morphology, endoskeleton and dentition are more detailed and accurate, the new figures provided are adequate and of high quality, reinforcing the results of the authors; and all the evidence support the conclusions of the authors.

For these reasons I recommend to publish the manuscript as it is.

Reviewer #3

(Remarks to the Author)

Briefly - I accept the revisions. The figures and description are improved significantly. I have no further contributions that I wish to offer that might delay the publication of this work.

Version 2:

Reviewer comments:

Reviewer #2

(Remarks to the Author)

After reviewing the manuscript, and having into account the response of the authors to the comments of reviewer 1, I believe that the authors have addressed in a satisfactory manner their concerns, and the manuscript can be published as it is.

R#1: Regarding the presence of only a single barbel in the fossils species, I agree with the authors that this seems a very characteristic feature of this species, so it is an important character that needs to be described. The diversity of chondrichthyans body plans we find in the fossil record is incomplete and imperfect. The presence of a single throat barbel in this species, although unusual it is not an outlandish claim, and in four of the five specimens studied the presence of this barbel is very clear (in the fifth is not possible due to its position), and consistent it is position and morphology, so I agree with the authors in the inclusion of this feature in the new amended diagnosis.

R#2: The authors acknowledge the claim of reviewer about the difficulty of assessing the state of development of the skeletons and the ontogeny stage of the specimens found and have amended their reference, presenting a new one that properly back up their point. Moreover, the authors prove that the small size of the specimens doesn't immediately correlate with juveniles stages of the ontogeny in fossil sharks, and provide references to support this claim.

R#3: Regarding the assignation of the isolated teeth to †Bavariscyllium tischlingeri, I agree with the authors in their attribution to this species. After reviewing the literature, the teeth figured in Thies 2005, both from the holotype and the isolated teeth assigned to †Bavariscyllium sp. have enough similarities with the material presented in this work to be assigned to the same species. It is also clear that the teeth attributed to †Thiesus concavus (Guinot et al. , 2014) present enough similar characteristics to be synonymised as †Bavariscyllium tischlingeri.

R#4: I think that the authors have approach the problems that integrating molecular and fossil data in phylogenies in a very cautious manner. They took into account the fragmentary nature of the fossil record and the morphological data that can be extracted from it, and their results reflect the uncertainty of the placement of the fossil taxa, which is a common occurrence in this type of studies.

Dr. Sebastian Stumpf
Senior Research Fellow
University of Vienna
Department of Palaeontology
Josef-Holaubek-Platz 2 (UZA II)
1090 Vienna, Austria

Vienna, 9 May 2025

Dear Editor,

Following the initial submission of our manuscript **COMMSBIO-24-7830**, entitled 'Reappraisal of the extinct barbelthroat shark †*Bavariscyllum* and the nebulous origin of carcharhiniform galeomorphs', it has been thoroughly evaluated by three peer reviewers. Their feedback was both insightful and constructive, offering valuable suggestions for enhancing the clarity, depth, and rigor of our work. We sincerely acknowledge their time and efforts in reviewing the manuscript, and we are grateful for the opportunity to address their points.

We have added a new co-author to this work, Andreas Hecker from the Jura-Museum Eichstätt in Germany, who contributed additional images of the holotype specimen of †*Bavariscyllum tischlingeri*, both under normal and UV light, in response to a suggestion from one of the reviewers. Most importantly, his UV light images facilitated the identification of previously overlooked teeth preserved in the holotype, which have now been incorporated into the revised manuscript, further strengthening the results of our study.

Sincerely,

Sebastian Stumpf & Jürgen Kriwet

on behalf of all co-authors

Reviewer #1

This well-written manuscript presents a study of fossil shark specimens that are exceptional for their quality of preservation and age. These specimens are key fossils for understanding the evolution of modern elasmobranchs and their initial radiation during the Jurassic period. While the results may not be totally novel they will be of interest to the paleoichthyological community for applying analytical methods. This manuscript is part of the ongoing work of this team, which aims to re-examine and revise the systematics of old specimens collected in Late Jurassic Solnhofen Archipelago in southern Germany (e.g. Kriwet & Klug, 2005; Lopez-Romero et al., 2020; Stumph et al., 2022; Jumbura et al., 2023), whether from public or private collections, which are now available to researchers. In this respect, the aim of this re-study of elasmobranch skeletons of enigmatical species *Bavariscyllium tischlingeri* using two analytical approaches (morphometry, phylogeny) is really exciting. However, the manuscript contains flaws and/or shortcuts that make the results questionable and the systematic review, in my opinion, unsatisfactory. The main criticisms are essentially due to flawed nested reasoning, namely:

- that (1), the authors partially revised diagnosis and systematic affiliation of *Bavariscyllium tischlingeri* from the Solnhofen region by first attributing teeth from another region and another age without any serious argument or study. The proposed new diagnosis of this species is based in particular on these dental characteristics observed on isolated teeth from another locality ("comprising hundreds of isolated teeth whose taxonomic affinities have yet to be fully established by Kriwet & Klug 2004; Klug 2009") collected in deposit of older age. The authors justify this systematic comparison in a few words by a global resemblance (unproven in my opinion) to the rare teeth of the holotype, as published by Thies & Leidner in 2011.

- That (2), the authors have excluded the possibility that these 4 semi-complete specimens (<25 cm) are juveniles, which would invalidate in part their comparative morphometric analysis (not verifiable due to missing data in supplementary data where measurements are hidden with #####) and their phylogenetic analysis.

- That (3), The authors do not really describe the skeletal anatomy of the specimens in any way. This makes it difficult to understand the coding of characters (phylogeny) and the main result (placement close to the Carcharhiniformes). We can only verify this by looking at the teeth, but unfortunately these do not belong to the holotype or the associated specimens (see point 1). I haven't been able to work out how the characters were coded from the figures as available in the MS.

I have listed below some questions/accuracies and the main points/arguments that I found problematic and need to be checked by the authors.

R#1-1: Anatomy

Barbels: why a single? why there is no pair of barbels as expected in sharks. Preservation artefacts resulting from lateral compression could erase the second one, note it is on the right in the three right flanked specimens and on the left in the left flanked specimen (and missing in the dorsoventral specimen)? Regarding the right/left position of the barbels, depending on the side of observation, we can assume that there are two, as for the pectoral and pelvic fins?

Response: As detailed in the Description section, the possibility that †*Bavariscyllium* may have possessed two throat barbels cannot be entirely ruled out, and therefore, the presence of a single throat barbel is considered provisional, pending substantiation by additional fossil evidence. Considering this context, we would prefer not to make changes to this part of the text.

R#1-2: Gender: Authors argued that all specimens are (?adult) female, lacking visible claspers on specimens. Why not all juveniles? as all specimens are small (<25cm TL) and that all specimens come from the lagoon area of Solnhofen. This point, which is completely overlooked, is crucial for the rest of the analysis. In fact, the analysis of juveniles becomes complicated, both morphometrically and phylogenetically, because it encounters problems of allometry that affect both biometric and phylogenetic criteria, especially since the living specimens used for comparison are all adults.

Response: Despite the small size and lack of claspers (potentially suggesting juvenile stages), the specimens are interpreted as representing subadult to adult individuals, based on the advanced mineralization of their endo- and exoskeletons, even though the endoskeleton of the smallest specimen (SMF P 272) appears less mineralized, indicating a potentially earlier developmental stage. This interpretation aligns with the prolonged skeletogenesis observed in extant elasmobranchs, which starts during late embryonic development and continues well beyond hatching, with well-mineralized claspers (used to determine maturity in males) usually developing during late ontogeny. The relevant section has been modified accordingly and now reads as follows:

The endoskeleton, where discernible, comprises elements of both the axial and appendicular skeleton. The endoskeleton of SMF P 272, while appearing fully developed (Figure 3C–E), seems less well-mineralized than that of the larger specimens, suggesting an earlier developmental stage. Despite the small size of the available specimens and the absence of claspers in any of them, which might suggest juvenile stages, they are interpreted as representing subadult to adult individuals. This interpretation is based on the apparently advanced mineralization of both their endo- and exoskeletons, in conjunction with the prolonged skeletogenesis observed in extant sharks, which starts during late embryonic development and continues well beyond hatching (e.g., Enault et al. 2015, 2016; López-Romero et al. 2022), with males usually developing well-mineralized claspers during late ontogeny, upon attaining sexual maturity (e.g., Braccini & Chiaramonte 2002; Lucifora et al. 2004; Jones et al. 2008).

R#1-3: Teeth: Attribution of several isolated teeth from another locality (Mahlstetten) and another age (here, Kimmeridgian) to this Tithonian taxon from Eichstätt (Solnhofen area) is extremely problematic for me, especially because they are used to modify original diagnosis, to code morphological characters and support phylogenetic analysis. This cannot be done because there is nothing to indicate with certainty that these teeth belong to this species (and also to this genus).

The authors avoid to discuss about this, considering in one short sentence that they resemble the holotype illustrated by Thies & Leidner (2011). However, they appear to be quite different from teeth belonging to holotype of *Bavariscyllum tischlingeri* Thies, 2005 (SOS 4124), from the Eichstätt Tithonian and well-illustrated by Thies & Leidner (2011: plate 45). The teeth figured by Thies & Leidner (2011) are of lesser quality, the roots of the teeth still being absent, but they have the merit of belonging to the holotype (extracted from jaws). The dentition of the holotype is much less heterodont than that shown by the authors from the dentition reconstituted from isolated teeth coming from Kimmeridgian of Mahlstetten deposit, the posterior teeth being less spread out. The cusps of the anterior and anterolateral teeth of holotype are much more slender, high and developed than those of the Kimmeridgian, and the heels never show well-formed denticles, whereas those of the Kimmeridgian show up to two pairs of well-detached and sometimes divergent secondary denticles. The main cusp is much more curved lingually in Tithonian teeth, whereas the Kimmeridgian cusps have a fairly flat labial surface. The ornamentation of the crown is much weaker on the teeth of the holotype compared to those of the Kimmeridgian. The roots are not known on the teeth of the holotype, so it is difficult to compare them with those of the Kimmeridgian. Even if the authors do not agree with my previous remarks, they cannot attribute isolated Kimmeridgian teeth to a Tithonian species without a more solid argument and, above all, use them to redefine the

species. To conclude, we cannot yet be sure that the Kimmeridgian teeth belong to this species. The Lower Kimmeridgian teeth from Mahlstetten are much more similar to species *Thiesus concavus* from Cretaceous than they are to *Bavariscyllum tischlingeri*, as Thies & Leidner (2011) figure. The authors discuss this last great resemblance, but later in the manuscript, by placing the cretaceous species *Thiesus concavus* as a synonym of Jurassic *Bavariscyllum tischlingeri*, omitting that the Kimmeridgian teeth could also belong to *Thiesus* and not to *Bavariscyllum*...

Actually, all the dental features used in the updated diagnosis and phylogenetical analysis (characters 199-212) need to be removed to be totally accurate and serious. Only the teeth described by Thies & Leidner (2011) can be safely said to belong to species, and only these can be used to identify, describe and used for coding *Bavariscyllum tischlingeri*.

Response: An evaluation of additional UV light images of the holotype specimen of †*Bavariscyllum tischlingeri*, conducted in response to a suggestion from one of the reviewers (see R#3-6), revealed the presence of previously overlooked lateral teeth, including one nearly complete specimen. Importantly, these teeth share key dental features with those from the lower Kimmeridgian of Mahlstetten and confirm, among other traits, the presence of a bilobate root with diverging labial lobes in the holotype of †*Bavariscyllum tischlingeri*. While we were already confident in our conclusion prior to obtaining these images, the new findings now provide definitive support for our assertion that the Mahlstetten teeth can be unambiguously referred to †*Bavariscyllum tischlingeri*.

However, it has come to light that the holotype specimen is coated with a very thin, transparent consolidant of unknown chemical composition, likely applied to seal and preserve the specimen. While macroscopic photography is not substantially impaired, the coating tends to blur fine microscopic details, although these features become considerably more discernible under UV light illumination. Capturing minute dental features, such as tooth crown ornamentation, nevertheless proved challenging; however, the overall morphology of the preserved teeth remains sufficiently identifiable to permit meaningful morphological assessment.

The description of the Mahlstetten teeth and the accompanying figure were revised and expanded, as also recommended by Reviewer #2 (see R#2-2), to facilitate clearer identification of the described morphological features. Teeth previously identified as anteriors are reclassified as laterals, which better aligns with the heterodonty observed in *Thiesus concavus* by Guinot et al. (2014), here treated as a junior synonym of †*Bavariscyllum tischlingeri*, with the morphological variation passing posteriorly through the dentition, primarily involving a reduction in the size of the main

cusps and the spreading of the labial root lobes. Details regarding the holotype's preservation and our microscopic imaging setup, including camera mount specifications, have been included in the revised manuscript to clarify our methodology. The revised sections read as follows:

Dentition. The dentition of †*Bavariscyllium tischlingeri* remains poorly understood, with previous information limited to four incomplete teeth extracted from the holotype specimen, which Thies (2005) identified as one anterior, one antero-lateral, one lateral, and one posterior tooth, all missing most of their roots. The exact origin of these teeth, whether from the lower or upper dentition, remains undetermined.

Our re-examination of JME SOS 4124 revealed the presence of four reasonably well-preserved lateral teeth including one that is nearly complete (Figure 2C, D), which appear to have been overlooked in the original description by Thies (2005). However, JME SOS 4124 is coated with an unknown transparent consolidant that is permeable to UV light, presumably applied to seal and stabilize the specimen and containing numerous embedded dust particles and fibers. This coating impedes the visibility of the teeth under normal light conditions, although they become considerably more discernible under UV light illumination. Nevertheless, the consolidant obscures minute morphological features, such as crown ornamentation. Despite these limitations, the overall morphology of the preserved teeth remains sufficiently identifiable to permit meaningful morphological assessment. As with the extracted teeth described by Thies (2005), the assignment of these additional teeth to either the lower or upper dentition remains uncertain. The teeth are visible in labial, lingual, and lateral views and exhibit a pointed main cusp flanked by one or two pairs of small lateral cusplets. Notably, the lateral cusplets appear slightly more developed than those in the extracted teeth described by Thies (2005), indicating a higher degree of heterodonty than previously recognized. Continuous cutting edges are present, running across the main cusp and lateral cusplets. Tooth crown ornamentation appears to be reduced and includes at least two faint oblique ridges extending along the labial crown face. The labial crown face exhibits a slightly concave basal edge, and the root is bilobate, with diverging root lobes protruding labially below the crown. The lingual protuberance of the root appears moderately well developed and is basally pierced by a single foramen.

The isolated teeth from the lower Kimmeridgian of Mahlstetten in southern Germany, which share key dental features with the holotype, provide additional morphological insights that help clarify the dental morphology of †*Bavariscyllium tischlingeri*.

The teeth from Mahlstetten are very small, measuring less than 1 mm in height, and can be separated into those coming from tooth files of antero-lateral, lateral, and posterior positions (Figure 5). Anterior teeth, which are the largest in the jaws and characterized by a very high and slender main cusp (Thies 2005), could not be identified. The teeth are generally well-preserved, but have suffered post-mortem damage in some places, with some teeth displaying patterns of bioerosion, likely caused by endolithic microorganisms (compare Underwood et al. 1999).

Antero-lateral teeth (Figure 5A–E) are slightly asymmetrical and higher than wide, with a high, pointed main cusp. Below the main cusp, the crown slightly widens, giving rise to a pair of distally diverging heels and resulting in a labial crown face with a concave basal edge that overhangs the root (Figure 5A). The lateral heels are labially displaced with respect to the main cusp, resulting in a flat or slightly convex labial crown face. Each heel bears two pairs of rudimentary cusplets, the most mesial and distal of which are barely discernible. There is neither an apron nor a uvula. The cutting edges are weak and continuous across the main cusp and lateral cusplets (Figure 5B). The labial crown face bears two oblique ridges that extend along the lateral heels up to the base of the main cusp. Additionally, a few straight to slightly undulating vertical ridges run along the main cusp and lateral heels. The lingual crown face exhibits a few vertical ridges that are confined to the lower half of the main cusp (Figure 5C). There is a well-developed enameloid collar that covers the uppermost part of the root along the entire crown-root junction.

The root is very low and flat with a flared basal face (Figure 5D). It is Y-shaped in basal view with a low, slightly swollen lingual protuberance and flared root lobes that protrude below the lateral heels of the crown. The root vascularisation is hemiaulacorrhize, with the lingual protuberance of the root pierced basally by a single foramen that connects to the central basal foramen by an internal canal (Figure 5C). Laterally, at the level of the first pair of lateral cusplets immediately below the collar, there is a single large foramen, accompanied by a series of smaller foramina extending below the second pair of lateral cusplets (Figure 5B, E).

Lateral teeth (Figure 5F–P), similar to antero-laterals, are higher than wide but symmetrical, with more well-developed lateral heels and cusplets, and more strongly flared labial root lobes that extend well below the lateral heels of the crown (Figure 5G, J, N). The labial crown face has a strongly concave basal edge and may be devoid of ornamentation (Figure 5N), but it usually exhibits two oblique ridges that ascend along the lateral heels up to the lower part of the main cusp (Figure 5G, J). From these ridges, weak vertical ridges may branch off, and straight to slightly undulating vertical ridges may occur on the main cusp, but do not reach its apex. The vertical ridges extending

lingually may reach the apex of the main cusp (Figure 5L, P). As seen in antero-lateral teeth, a single large foramen and an accompanying series smaller foramen may be present, penetrating the root laterally immediately below the lateral cusplets (Figure 5H, I). Labially, the root may be perforated by a row of foramina below the collar where the lateral root lobes join (Figure 5M).

Lateral teeth from more posterior positions (Figure 5R–U) are asymmetrical, displaying a distally inclined main cusp and a slightly more elongate mesial crown heel and root lobe (Figure 5R, T). The lateral cusplets are reduced and may be barely discernible (Figure 5R).

Exhibiting the characteristic ornamentation of lateral teeth, posterior teeth (Figure 5V–C') are about as high as wide, with a short but stout main cusp and typically up to two pairs of lateral cusplets. The mesial heel may bear three pairs of lateral cusplets, the most mesial one being very minute and barely visible (Figure 5X). No foramina perforating the root could be observed.

Discussion

[...]

The teeth from the lower Kimmeridgian of Mahlstetten, which share characteristic dental features with the holotype of †*B. tischlingeri* (including a concave labial crown edge and a pair of labially displaced lateral heels, each with small cusplets and typically an oblique labial ridge, as well as divergent labial root lobes and a weak lingual protuberance pierced basally by a single foramen), provide valuable new insights into the species' dental morphology. These scyliorhinid-like teeth, characterized by a low hemiaulacorhize root exhibiting strongly flared lobes, closely resemble †*Palaeoscyllium tenuidens* from the Bathonian of England (Underwood & Ward 2004) and †*P. formosum* from the Kimmeridgian–Tithonian of Germany, France and England (Candoni 1993; Underwood 2002; Kriwet & Klug 2004). The closest similarities are with †*Thiesus concavus* from the Valanginian of France (Guinot et al. 2014), with some teeth being nearly indistinguishable, differing from †*T. concavus* only in having main cusps that are slightly more robust and less lingually bent. However, this very minor difference, along with the shared lack of labial ornamentation in some teeth, is more likely to reflect intra- rather than interspecific variation, such as gynandric and/or ontogenetic heterodonty, as documented in many extant elasmobranch species (e.g., Gutteridge & Bennett 2014; Underwood et al. 2015; Cullen & Marshall 2019; Ebersole et al. 2023) and particularly well expressed in carcharhiniforms with generalized clutching-type dentitions (Herman et al. 1990; Soares & de Carvalho 2019; Berio et al. 2020, 2022). Consequently, due to the striking similarities between the Mahlstetten teeth and †*T.*

concavus, we propose treating the latter as a junior synonym of †*B. tischlingeri*, extending its stratigraphic range into the Early Cretaceous.

Material and methods

[...]

Previously undocumented teeth of the holotype specimen of †*Bavariscyllium tischlingeri* (JME SOS 4124) shown in Figure 2C, D, are coated with an unknown transparent consolidant, likely applied to seal and stabilize the fossil, and were photographed under 365 nm UV light using a Canon EOS 800D DSLR camera mounted with a LIUZHENZHEN NDPL-2 microscope objective on a Novex Trino Zoom. The resulting images were post-processed in Adobe Photoshop CC 2021 for improved clarity, without altering morphological details.

R#1-4: Morphometry

As mentioned above, it is quite possible that the four Solnhofen fossils are juveniles, because they are very small (less than 25 cm) and the way the area used to be (the palaeoenvironment) suggests that it was a lagoon, which is a great place for young animals to grow up. Unfortunately, the measurements of the fossils (and those scanned from Book of Ebert, 2021) are not available in the supplementary data. I cannot check for a possible effect of size and/or allometry. Whatever, this possibility to have only juveniles is never discussed in MS.

Response: In view of the inferred subadult to adult ontogenetic stages of the specimens (see R#1-2), a 24% size difference between the smallest and largest specimen, and the results of our morphometric analysis, the overall body shape of †*Bavariscyllium* appears to have been conserved throughout ontogeny. However, this conclusion remains tentative pending further testing with a larger sample size. To provide greater clarity, we have expanded on the relevant sections, including the Description and Morphometrics sections, as follows:

Description

[...]

The body of †*Bavariscyllium* is elongate and slender, with a total length ranging from approximately 190 mm (SMF P 272) to 250 mm (SMNS 96086), reflecting a 24% variation in total length between the smallest and largest specimen.

Morphometrics

[...]

By contrast, and despite a maximum size difference of 24% between the smallest and largest specimen of †*Bavariscyllium*, the taxon displays a distinct aggregation, closer to carcharhiniforms (more precisely pentanchids) than to orectolobiforms. This distinct morphospace occupation suggests that overall body shape of †*Bavariscyllium* may have been conserved throughout ontogeny, although this interpretation requires further testing with a larger sample size.

R#1-5: I also wondered why *Bavariscyllium tischlingeri* is compared only with the coeval species *Paleoscyllium formosum*, and not with others such as the coeval orectolobiform "*Crossorhinus jurassicus*", the dubious "*Palaeoscyllium minus*" from Eichstätt (see Thies & Leidner, 2011), or the possible orectolobiform *Paleocarcharias stomeri* from Blumenberg also studied by the same team? It would make sense to use both a more probable carcharhiniform like *Paleoscyllium formosum* and a potential orectolobiform from these same deposits/ages to compare *Bavariscyllium* from elements that have undergone comparable deformation/compression.

Response: As detailed in the Discussion section, the inclusion of fossil taxa with less streamlined and more deep-bodied body plans, such as †*Palaeocarcharias*, would certainly expand the morphospace and provide additional context for Jurassic shark disparity. However, our study focused on †*Bavariscyllium* and †*Paleoscyllium*, as both were originally referred to Scyliorhinidae based on their close dental resemblance to extant species traditionally included in this family. While a broader comparative approach could yield further insights, particularly regarding ecological overlaps with extant taxa, it falls beyond the scope of this study and will be addressed in more detail in a forthcoming work.

R#1-6: Even unnecessary with PCA based on covariance, are the different sizes normalised by TL? If not normalised by TL, the large variance induced by a large heterogeneity in comparative sample could minimise some effects on small specimens such as *Bavariscyllium tischlingeri*.

Response: Yes, they are normalized by TL. This normalization ensures that differences in overall body size do not drive the patterns observed in our analysis. By using relative measurements, we account for potential size-related variation, allowing for more accurate comparisons between specimens, including smaller ones of †*Bavariscyllium*.

R#1-7: The authors suggested ontogenetic allometry for *Palaeoscyllium formosum* because the large separation between the two *Palaeoscyllium* specimens is intriguing and could be due

to ontogenetic allometry, as has been documented in extant sharks (e.g. Ahnelt et al. 2020; Sternes & Higham 2022; Gayford et al. 2024). This is certainly possible, and it would have been interesting if the authors had asked the same questions about the target species *Bavariscyllium tischlingeri*.

Response: As outlined above, current evidence does not suggest allometric growth in †*Bavariscyllium*.

R#1-8: The authors successfully tested which measurements were normally and non-normally distributed using a variety of statistical approaches. However, it is not clear why they do not use only these last characters to study the position of their two fossil taxa. We can expect that measurements normally distributed probably covariate as the whole. The authors also used morphological disparity (Figure 5), Can they specify what they are analysing? covariance matrix divided by the number of observations in the group?

Response: To ensure that all relevant morphological information was retained, we included both normally and non-normally distributed variables in our analyses. Excluding non-normally distributed variables could lead to the loss of important variation, which is particularly relevant for fossil taxa with limited sample sizes. While normally distributed measurements may covary, this is not always the case, as shown by negative correlations between most measurements (Figure R1).

Figure R1: Correlation plot between all the variables used in the study. Abbreviations follow the same convention as used in the Supplementary Information.

An important example is PD1 (pre-first dorsal length), which is not normally distributed but highly significant for our study. Thies (2005) assigned †*Bavariscyllium* to Scyliorhinidae based on the position of the first dorsal fin relative to the pelvic fins. Given its importance, excluding PD1 solely due to its distribution would have led to the loss of a crucial morphological character for evaluating the placement of †*Bavariscyllium*.

Additionally, in our PCA, we set `.scale = FALSE` in the `prcomp` function (previously `.scale = TRUE`). This prevents variance in our dataset from being artificially influenced by differences in trait variability. Since all our measurements were standardized as percentages of TL, further scaling was unnecessary and could have distorted the biological signal by disproportionately weighting traits with initially lower variance. By not scaling, the PCA better reflects true proportional differences in morphology rather than overemphasizing traits with naturally lower variability. However, the overall results remain consistent, with only slight alterations, reinforcing confidence in our analysis.

We performed the disparity analysis using all 15 PC scores and estimated it based on the sum of variances. Applying the covariance matrix divided by the number of observations, as implemented in the `morphol.disparity` function of *geomorph*, yields similar results.

```
morphol.disparity;  
Carcharhiniformes = 162.4049  
†Bavariscyllium = 12.3165  
†Palaeoscillyum = 27.6225  
Orectolobiformes = 192.4136
```

```
dispRity.per.group  
Carcharhiniformes = 163.4263  
Bavariscyllium = 16.422  
Palaeoscillium = 55.2451  
Orectolobiformes = 200.4308
```

We chose to present the results from the `dispRity.per.group` function of *dispRity*. This approach is well-suited for comparing subgroups within the same morphospace, as the major axis of variation for the entire group may not align with that of the subgroups

(Hopkins and Gerber 2017). Additionally, we used the sum of ranges as an alternative measure of disparity, which is more sensitive to outliers and variations in sample size.

Hopkins, M.J., & Gerber, S. 2017. Morphological Disparity. In Nuno de la Rosa, L., & Müller, G., (eds.), *Evolutionary Developmental Biology*. Springer, Cham.

The Morphometrics and M&M sections were adjusted accordingly, as follows:

Morphometrics

To quantify the body shape of †*Bavariscyllium tischingeri*, a set of 16 linear body measurements was used (Supplementary Figure S1) to compare this species, along with †*Palaeoscyllium formosum*, to extant demersal sharks with elongate, fusiform precaudal body designs, comprising species in the carcharhiniform families Scyliorhinidae, Atelomycteridae and Pentanchidae, and phenotypically similar sharks in the orectolobiform families Parascylliidae and Hemiscylliidae (Figure 6A; Supplementary Table S1). The compiled dataset was subjected to a principal component analysis (PCA), resulting in 15 axes (Supplementary Table S2). The first three axes each explain more than 5% of the variation and together account for 89.51% of the total variability (see Supplementary Table S2 for information on all PC axes, and Supplementary Table S3 for loadings of the first three PC axes). Plotted on PC1 (55.04% of total variation) and PC2 (24.3%) the morphospace occupations in Figure 6B shows hemiscylliids distinctly separated from all other taxonomic groups, occupying negative PC1 and primarily positive PC2 values. Pentanchids are widely scattered and occupy the largest morphospace within the dataset, spanning both positive and negative values of PC1 and PC2, although they tend to cluster around positive PC1 and negative PC2 values. There is a considerable overlap between pentanchids and scyliorhinids, with the latter predominantly occupying negative values along PC2, showing only a narrow extension into positive PC2 values, and spanning both positive and negative values along PC1. Parascylliids partially overlap with both pentanchids and atelomycterids and are mostly restricted to negative values along PC1 and PC2. Atelomycterids exhibit a partial overlap with pentanchids, scyliorhinids, and parascylliids, occupying predominantly negative values along both axes. †*Bavariscyllium* (n = four specimens) and †*Palaeoscyllium* (n = two specimens) are distinct from one another, with †*Bavariscyllium* occupying a unique morphospace that does not overlap with any extant family. One specimen of †*Palaeoscyllium* overlaps with parascylliids, atelomycterids, and pentanchids, while the other falls outside the morphospace of any living family. Grouping extant species at order level reveals an

overlap of †*Palaeoscyllium* with both orectolobiforms and carcharhiniforms (Figure 6C). By contrast, and despite a maximum size difference of 24% between the smallest and largest specimen of †*Bavariscyllum*, the taxon displays a distinct aggregation, closer to carcharhiniforms (more precisely pentanchids) than to orectolobiforms. This distinct morphospace occupation suggests that overall body shape of †*Bavariscyllum* may have been conserved throughout ontogeny, although this interpretation requires further testing with a larger sample size.

Plotting PC1 against PC3 (10.17% of total variation) reveals a similar morphospace occupation (Supplementary Figure S3). Hemiscylliids are distinctly separated, occupying exclusively negative values along both axes. Pentanchids exhibit the largest morphospace occupation, spanning both positive and negative values of PC1 and PC3. Scyliorhinids nearly completely overlap with pentanchids, occupying both positive and negative values of PC3 and predominantly positive values of PC1. Parascylliids partially overlap with pentanchids and atelomycterids, restricted to negative PC1 and positive PC3 values. Atelomycterids extend across both positive and negative PC3 values and are almost exclusively negative along PC1, with partial overlap with parascylliids and pentanchids, and minor overlap with scyliorhinids. †*Bavariscyllum* and †*Palaeoscyllium* are distinct. †*Bavariscyllum* overlaps with pentanchids in the positive regions of PC1 and PC3, while one specimen of †*Palaeoscyllium* overlaps with parascylliids and atelomycterids in the negative region of PC1 and positive region of PC3. The second †*Palaeoscyllium* specimen plots near Parascylliidae.

The relatively wide separation between the two †*Palaeoscyllium* specimens is intriguing, but given the substantial size difference between them (Supplementary Figure S2), it may be attributed to ontogenetic allometry, as has been documented in extant sharks (e.g., Ahnelt et al. 2020; Sternes & Higham 2022; Gayford et al. 2024). One of the most striking feature distinguishing †*Palaeoscyllium* from †*Bavariscyllum* (and any other taxon included in the present dataset) is the position of the first dorsal fin, which originates anterior to the pelvic fins (Figure 6D). This condition, along with a slightly more robust precaudal body and a truly heterocercal caudal fin with a distinct ventral lobe, indicates a closer resemblance to †*Corysodon* (Saint-Seine 1949; Thies & Candoni 1998) and thus to sharks of the littoral rather than the leptobenthic ecomorphotype (Compagno 1990; White et al. 2022).

A Shapiro-Wilk test showed that 46.67% of all measurements are normally distributed (i.e., seven out of 15 measurements; Supplementary Table S4). A Kruskal-Wallis test revealed significant differences at family level for all non-normally distributed measurements (Supplementary Table S5) and significant differences at order level for

all but one non-normally distributed measurement (i.e., interdorsal space [IDS]; Supplementary Table S9). Significant differences were also found at both family and order levels when pairwise Wilcoxon tests were used (Supplementary Tables S6 and S10). ANOVA tests on each normally distributed measurement demonstrated significant differences at family and order levels (Supplementary Tables S7 and S11). These findings also are corroborated by the results of the pairwise comparisons (Supplementary Tables S8 and S12 for comparisons at family and order levels, respectively).

The dataset was further subjected to a linear discriminant analysis (LDA) to explore the separation between the *a priori* groups using the leave-one-out cross-validation (LOOCV) approach. The results indicate that neither †*Bavariscyllium* nor †*Palaeoscyllium* can be phenotypically allied with any of the analyzed families (Supplementary Table S13), with a LOOCV accuracy of 93.2%. When extant species are grouped at order level, it becomes evident that †*Bavariscyllium* is distinct from both carcharhiniforms and orectolobiforms, with a LOOCV accuracy of 96.3%, while †*Palaeoscyllium* is found to be allied with carcharhiniforms (Supplementary Table S14). A similar picture emerges when the morphological disparity, calculated as the sum of variances, is considered, further highlighting the distinct nature of †*Bavariscyllium* (Figure 6E; Supplementary Tables S15 and S16). The wide disparity displayed by †*Palaeoscyllium* aligns with the results from the PCA, but a larger sample size is needed to draw any conclusions. Overall, †*Bavariscyllium* exhibits a variance distinct from all comparative groups. Disparity calculated as the sum of ranges yields very similar results to those from the sum of variances (Supplementary Tables S17 and S18). Pairwise comparisons at order level remain consistent, while the family-level subsets show slight changes, particularly with differences between †*Bavariscyllium* and †*Palaeoscyllium*.

R#1-9: Phylogeny

I am a bit surprised that authors do not have a better resolution with such large matrix, whereas Vullo et al., 2024 have a well resolved strict consensus for the Galeomorphi with 6349 trees? For instance, why you remove some fossils from the same deposits (e.g. stem Squalomorphii Protospinax) and not others (stem Galeopmorphii Paracestracion)? Is there a reason for this omission? The matrix of Vullo et al. 2024 is in turn taken from Jamboura et al. 2023, which in turn is taken from Landemaine et al. 2018 and Villalobos-Segura et al. 2022 etc,... with some new modifications by Authors that do not specifically concern the requirements for the study of target species (and coded as ?)...? Why? On the other hand, the presence/absence of barbels, a character much emphasised by authors on fossil material, is not even used in the

matrix, Why ???? It could possibly give an advantage for a placement with the Orectolobiformes that possess such feature.

Regarding the synapomorphic characters of the clade (Carcharhiniformes and Bavariscyllium), I have noticed that very little has been documented on fossil taxa, suggesting that the groupings are essentially based on homoplasy :

- e.g. characters 17[0], 33[4], 123[1], 174[1] (synapomorphy of Carcharhiniforme + Bavariscyllium), only character 123[1] is really coded (presence of a Scapular_posterior_process) for Bavariscyllium. Bavariscyllium is therefore only related to Carcharhiniformes on the basis of homoplasy and the presence of a posterodorsal process at base of scapular process (pd). Da Silva et al. (2017) suggested that presence of two posterior triangular process of scapula (pdt and pvt) could be characteristic of "stem" Carcharhiniforms. They hypothesized that the posterior ventral (pvt) and dorsal (pdt) triangular processes found in Scyliorhinidae, Triakidae and Proscylliidae are multiple independent acquisitions within Carcharhiniformes. Have you seen these peculiar triangular processes on the four specimens?
- or e.g. characters 3[2], 9[0], 114[0], 135[0], 203[0] (synapomorphy of the clade (Carchar+Lamnif+Bavariscyllium+Palaeoscyllium), so that only character 203[0] = orthodontine, is coded on the teeth supposed to belong to Bavariscyllium (see previous chapter).

To conclude, the position of Bavariscyllium close to the Carcharhiniformes (also previously attributed to this Order by formal author Thies, 2005) known from Tithonian skeletons (including the holotype), is based on an ambiguous character (123[1]) or on teeth that come from another locality and age rather than on teeth from holotype.

I am aware of the difficulty of studying fossil specimens, and that the anatomical mastery of authors is much greater than my own, but it is not possible to judge these characters coded in this way in the absence of sufficiently precise illustrations of the SNSB-BSPG 1878 IV 6, which nevertheless seems to have provided the maximum data. To complete my questioning, and after deciphering the matrix in the appendices : Bavariscyllium
???0????????????????001????????01????????01?0?1??0?????????1?????????11??110
11?????00?????1??11?2??00?????1?????0?0???10100000100---
34213???001?????01??0?11?1-11000?10010210000?????????????0---10000----
0100000?0000001

I have attempted to visualise the characters thus observed and coded for Bavariscyllium tischlingeri and I can't see/understand on the images provided how certain characters could have been coded (or possibly inferred) on these specimens whose skeleton pieces are not very well preserved (unlike the body print)? Do you have any other elements and/or other images that would make it possible to see and encode these characters? such as character 4

[0] that concerns the internasal plate, 21-22 (Interorbital_space / Optic_pedicel), 30 (Occipital_crest), 31(Basicranial_processes), 41 (Lateral_otic_process), 42(Postotic_process), 44 (Stapedial_arch), 49 (Metotic_otic_occipital_fissure), 59, 68-69 (e.g. Glossopharyngeal_nerve_path).... Only obvious skeletal/body print characters (such as 189-193 : Placoid_scales / Malar_and_alar_thornes / Lateral_rostral_dermal_denticles / up_Dorsal_fin_spines / Dorsal_Cephalic_spines ; 198-212 : Serrated_tail_sting) appear to be easily codable on these specimens. In cases where fossils are too uninformative/uncodable, why not use fewer characters (less ambiguous and/or observable) but apply a molecular backbone constraint to the extant taxa in previously published phylogenies?

Response: While the performance of all phylogenetic methods—particularly Bayesian inference and Maximum Likelihood—improves with dataset size, a larger matrix does not necessarily result in a more resolved tree. Moreover, a more resolved tree does not guarantee a correct answer. In parsimony-based analyses, resolution depends on the congruence among all included characters—both those that directly affect the target taxa and those that do not—since all characters in the analysis are considered putative synapomorphies.

In our study, poor resolution, particularly in analyses including both †*Palaeoscyllium* and †*Bavariscyllum*, is addressed. Regardless of †*Palaeoscyllium*'s inclusion, the resulting unresolved arrangement for these taxa reflects the limited morphological information available. As a result, we recommend treating both †*Bavariscyllum* and †*Palaeoscyllium* as *Galeomorphii incerti ordinis* and *incertae familiae*.

While our analysis does recover a relationship between †*Bavariscyllum* and carcharhiniforms, the high number of missing characters prevents confident support, as stated in the manuscript. As such, this relationship is considered one of several scenarios and remains tentative pending the discovery of more informative fossil material. †*Bavariscyllum* is placed in a polytomy with modern carcharhiniforms (e.g., *Rhizoprionodon*, *Sphyrna*, *Carcharhinus*, *Mustelus*, *Atelomycterus*, *Cephaloscyllium*, *Schroederichthys*, and *Scyliorhinus*), with weak support (Jackknife score of 20, bootstrap value of 1). We have added node support to Fig. 5 for clarity. Only one character (123[1]), a scapular with a posterior process, supports a more exclusive relationship, equivalent to the posterodorsal process described by da Silva et al. (2017).

Justification for taxon sampling

Any conclusion drawn from phylogenetic analysis depends on the selection of taxa. Our taxon selection primarily focuses on galeomorphs (comprising 30% of the taxa included

in the matrix), while also incorporating †*Palaeoscyllium*, as it has also been interpreted as representing a carcharhiniform galeomorph.

However, it is important to consider that a broader phylogenetic context is necessary for phylogenetic studies. An extensive outgroup is essential, as it not only allows testing the initial hypothesis—that †*Bavariscyllum* and †*Palaeoscyllium* are galeomorphs—but also enables evaluation of other possible relationships. The removal of taxa like †*Protospinax* is intended to reduce noise in the dataset and improve tree resolution by removing "wild card taxa" associated with other clades that are of no particular interest to the present study.

To summarize, our taxon sampling strategy was designed to provide a robust phylogenetic context, focusing on galeomorphs while incorporating relevant outgroups and removing destabilizing taxa.

Character coding and homology considerations

Regarding the comments on the sampling of morphological characters, we did not decide to retain a large amount of missing data or limit our coding to only easily observable characters. A phylogenetic revision is not merely about adding to previous frameworks, but about critically reviewing how characters were scored and which features were included. While it may be easier to rely on previous studies without reassessing their conclusions, a thorough revision involves re-examining both new material and past research to integrate new insights.

This study goes beyond the targeted taxa, revising 69 taxa in the matrix to establish a phylogenetic context for †*Bavariscyllum* and †*Palaeoscyllium*. The analysis includes a comprehensive sampling of anatomical features that define major elasmobranch groups, both extinct and extant. These groups are recovered in the analysis, with taxa classified within them. As such, this sampling provides a phylogenetic context for the fossil taxa under investigation.

While some characters, like the internasal plate, metotic otic occipital fissure, glossopharyngeal nerve path, occipital crest, and stapedia arch, might be difficult to observe and may not directly affect the targeted taxa, their inclusion and possible modifications are a result of this study's review. Though they may not be directly relevant to the target taxa, their inclusion supports other elasmobranchs where the fossil taxa could be placed.

On the other hand, characters such as the presence or absence of barbels raise homology issues that cannot be resolved with fossil material. Goto et al. (1994) established that the throat barbels in *Cirrhoscyllium* are not homologous to any other appendages in related shark groups based on internal morphology. The position of the

barbel in †*Bavariscyllium* could suggest cranial innervations similar to *Cirrhoscyllium*. However, this remains an assumption and cannot be verified. Given these issues, we chose not to include this feature in the phylogenetic analysis due to insufficient support for homology with *Cirrhoscyllium*.

Discussion of phylogenetic constraints

It can be argued that a sense of phylogenetic context could be provided by enforcing structure into a topology (backbones). However, this does not resolve the ambiguity associated with characters, nor is ambiguity directly related to how observable a character is. While forcing molecular backbones (topological constraints) is a valid approach, its effectiveness depends on the objective of the analysis: If the goal is to evaluate different evolutionary scenarios or estimate node ages, constraints can be useful. However, when studying phylogenetic relationships, the use of such constraints (particularly for the groups of interest) would not provide additional information that an unconstrained analysis would yield. While constraints may result in a more resolved topology and potentially place the target fossil taxa within modern groups, the character-based discussion of placement remains unchanged and would still need to be performed on the unconstrained tree, which shows the classification hypothesis based on the characters included in the analysis. Therefore, in phylogenetic studies, the use of topological constraints may obscure phylogenetic uncertainty within the dataset.

Despite these considerations, we agree with the comments regarding the presence of homoplasies in the present analysis, as observed in the characters supporting the relationship between †*Bavariscyllium* and †*Palaeoscyllium* with Galeomorphii (e.g., Char. 167 [2]—lack of a basal cartilage supporting the radial elements of the dorsal fins; Char. 192 [0]—lack of dorsal fin spines). While these characters are widely distributed among taxa in the present analysis, occurring in galeomorphs, batomorphs, and hexanchiforms, †*Bavariscyllium* and †*Palaeoscyllium* lack any synapomorphy for batomorphs or hexanchiforms. For example, †*Bavariscyllium* and †*Palaeoscyllium* do not exhibit a synarcual (fusion of cervical vertebrae), a defining feature of batomorphs. While hexanchiforms (e.g., *Heptanchias* and *Hexanchus*) also lack a basal element in their dorsal fin, they possess only a single dorsal fin, a synapomorphy for that clade. In contrast, †*Bavariscyllium* and †*Palaeoscyllium* present two dorsal fins.

Here is a brief explanation of the character scoring in response to the reviewer's comment: An internasal plate separating the palatoquadrates is considered a shared feature among many Palaeozoic chondrichthyans, while modern elasmobranchs,

including †*Palidiplospinax*, lack this feature, suggesting it was lost prior to the Jurassic. Given the presence of other modern elasmobranch features in both †*Bavariscyllium* and †*Palaeoscyllium*, we score this feature as absent. The same approach was also applied, e.g., to the optic pedicel, the lateral otic process, dorsal cephalic spines, and the postotic process. Features, such as the presence of spines preceding the dorsal fins, a feature found in some modern elasmobranchs as well as in several extinct relatives, including hybodontiform shark-like chondrichthyans, are clearly absent in †*Bavariscyllium* and †*Palaeoscyllium* and were coded as such. The examined specimens showed no anatomical evidence of a metotic-otic occipital fissure, which in modern elasmobranchs forms an embryonic space between the otic capsule and hypotic lamina and typically closes during ontogeny, leaving a small canal for the glossopharyngeal nerve. The corresponding characters were coded accordingly.

Reviewer #2

The manuscript title "Reappraisal of the extinct barbelthroat shark †*Bavariscyllium* and the nebulous origin of carcharhiniform galeomorphs" is a very interesting work in which the author review 5 articulated specimens and 17 isolated teeth belonging to the genus †*Bavariscyllium* in order to review the diagnosis of this genus. They also conducted some morphometric and phylogenetical analyses to test if this genus belongs to Carcharhiniformes or Orectolobiformes, and to try to elucidate early galeomorph evolution. Despite their extensive study, the authors couldn't assign this genus to any of the orders mentioned above, leaving them as *Incerti ordinis* and *Incertae familiae*. They stress the need for more research in order to untangled the placement of †*Bavariscyllium*, †*Palaeoscyllium*, †*Corysodon*, and some other species with uncertain affinities in the phylogeny and the early evolution of galeomorph sharks. Research like this is very important, because as the authors have stated, the great variability of dental traits in sharks sometimes cannot be use to infer phylogenetical placement, as convergent evolution could be the more parsimonious explanation for observed similarities. The inclusion of morphological data, although scarce in the fossil record, is very valuable for current and future research in which more suitable fossil material can be used in combination with the dental characters.

Overall, the authors have presented a well-researched work, an provided all the necessary data, and information about the software and packages used to replicate the results they provided. The manuscript is well-written, the quality of the figures is excellent and support most of the claims the authors made in the main text. The graphs are readable and informative, and the statistical analysis are adequate and support their conclusions.

R#2-1: Despite all the expose above, I have some comments and suggestions that I would like to ask the authors to consider. I have listed some of the most important here, but there are some more comments in the PDF attached:

Response: We carefully reviewed the annotated version of the manuscript and made modifications in accordance with the reviewers' comments.

R#2-2: 1. My main disagreement with the authors is the description of the isolated teeth. The authors state that "additional third pair of very small rudimentary lateral cusplets may be present" (lines 280-281), and also that "below the collar where the lateral root lobes join, there is a row of small foramina, and additional foramina may extend laterally along the length of the lateral heels" (lines 296-298). None of these features can be seen in the pictures provided by the authors: only two pair of lateral cusplets is depicted in all the teeth (with exception of the Fig. 3 G-H, that only shows one pair); and although most of the teeth preserve the root, I cannot distinguish the row of small foramina that the authors describe. Could the authors provide images in which this features appear, or highlight these features in the pictures already provided?

Response: We have updated the dental description and accompanying figure illustrating the teeth from Mahlstetten to facilitate clearer identification of described morphological features (see also R#1-3). In order to highlight diminutive lateral cusplets, which are difficult to discern, we have added arrows to the figure to indicate them.

R#2-3: 2. Line 230: I suggest to change the sentence "the splanchnocranium is visible in one specimen" to "the splanchnocranium is visible in specimen XXX"

Response: Thank you for your suggestion. We have revised the text accordingly.

R#2-4: 3. Line 311: Please, specify in which figure of Thies & Leidner (2011) are figured the dermal denticles. This is a very long paper with multitude of pictures, so to facilitate the readers to locate those dermal denticles more quickly.

Response: Thank you for your suggestion. We have revised the text accordingly.

R#2-5: 4. In my opinion, Figure 4A doesn't add much information in its current format, as all the information is in the supplementary material. For that reason, I suggest the authors to: a) put Supplementary Figure 1 as Figure 4A, or b) delete completely Figure 4A and refer only to Supplementary Figure 1.

Response: We have incorporated the change as recommended.

R#2-6: 5. In general, I recommend to the authors to cite more their figures in the whole description section. Please, some examples can be found in the annotated PDF attached.

Response: We have incorporated the changes as recommended.

R#2-7: 6. Some of the references are missing for the main text or the references section. There are also some discrepancies in the dates for some of them. Please, see the comments in the attached PDF.

Response: We have carefully cross-checked all in-text citations and references, corrected the missing entries, and resolved the discrepancies in publication dates as noted in the annotated PDF.

It is for all of the above comments, that I recommend some minor revisions to improve the manuscript before it can be published

Reviewer #3

Key results:

The authors present a re-description of the Jurassic elasmobranch shark *Bavariscyllium*. The revision has implications for how we understand the evolutionary radiation (morphospace occupation, time-tree node calibration) of major living groups – lamniform and orectolobiform sharks, although placement of this genus within the lamniform-carcharhiniform group is uncertain/unstable. More informatively, the body-shape and dentition indicate early ecomorphological specialization within the clade.

Originality and significance: The paper is a significant addition to a rapidly improving dataset documenting the historical biodiversity of a major division of extant vertebrates. Associating carefully described and quantified holomorphic fossil taxa with the vast record of isolated shark teeth is a significant addition in itself -testing tooth-based diagnoses and attendant hypotheses

of spatial-temporal taxon ranges. The careful treatment of phylogenetic output, coupled with morphometric analyses, are rarely seen applied to Mesozoic sharks.

Data & methodology: No problems – standard methods employed.... Matrix and new characters listed, supplied, and justified (a refreshing change).

Conclusions: See above.

Suggested improvements: See the specific comments below. Most importantly – add line drawings to explain/guide interpretation of skeletal remains in photographs. Pay more attention (in text) to details of appendicular and axial skeleton. Add a skeletal reconstruction, insofar as one can be completed. See questions raised about use of data lodged in private/semi private collections.

R#3-1: Consider changing the title! 'The nebulous origin...' kills interest - it suggests you've found nothing very much, which is not true.

Response: Thank you for your suggestion, but we are refraining from changing the title, as we believe it effectively conveys the narrative of our findings.

R#3-2: Specific comments:

L.109. Is it possible to show the former/current membership of Scyliorhinidae in Figure 6? number the figures for ease of identification

Response: The suggested change has been implemented.

R#3-3: L.121. Barbels as sense organs – additional information on kinds of sensory receptors known to be present – chemosensory?

Response: We have expanded on the relevant section, which now reads as follows:

As such, †*Bavariscyllium* appears to be readily distinguished from all living sharks. The latter commonly possess a bilateral pair of barbels associated with their nostrils (Ebert et al. 2021), innervated by cranial nerves (Shirai 1992) and serving as sensory organs for prey detection and environmental sensing through electroreception and possibly mechanoreception (Nevatte et al. 2017), although they do not appear to play a role in chemoreception (Fox 1999). An exception is the extant bottom-dwelling shark *Cirrhoscyllium*, comprising three northwestern Pacific species and placed within the galeomorph order Orectolobiformes and the family Parascylliidae (Goto 2001;

Compagno 2002). It is the only known shark to possess, in addition to short nasal barbels, a unique pair of whisker-like barbels originating from the ventral surface of the throat, likely serving a mechanosensory function (Goto et al. 1994).

R#3-4: L.151-174 – Diagnosis for genus lengthy, but most notable for more than half of its content characterizing details of the teeth. Does this indicate a largely tooth-determined phylogenetic placement? Can the diagnosis be condensed to include the most diagnostic features of the teeth?

Response: The diagnosis has been condensed and reads now as follows:

Revised diagnosis. Small-sized galeomorph shark characterized by the following combination of morphological characters: slender and elongate precaudal body; head short with a single whisker-like throat barbel; pectoral fins relatively small, aplesodic and rounded in shape; scapular process pointed with postero-dorsal process at its base; pelvic fins small, aplesodic and broadly angular to rounded, originating at about one-third of the total length; two relatively large dorsal fins with rounded tips lacking spines and basal plates, with the first dorsal fin originating posterior to the pelvic fin insertions; second dorsal fin equal to or slightly larger than the first dorsal fin; interdorsal space slightly wider than the dorsal fins are long; caudal fin low and very long, accounting for about one-third of the total length, with a subterminal notch but no ventral lobe; vertebral centra cyclospindylid; teeth with well-defined main cusp and a pair of lateral heels, each bearing no more than three, usually two, reduced lateral cusplets; labial crown face concave along its basal edge and overhanging the root; apron and uvula absent; cutting edges weak and continuous; labial crown face either devoid of ornamentation or with two oblique ridges extending along the lateral heels from which weak vertical ridges may branch off; well-developed enameloid collar along crown-root junction; low Y-shaped root with flat basal face and strongly flared labial root lobes; root vascularisation of hemiaulacorhizid type; lingual protuberance of the root pierced by single foramen, connected by an internal canal to central basal foramen; morphological variation passing posteriorly through the dentition involves a reduction in main cusp size and spreading of labial root lobes.

R#3-5: Figure 1 – clear, strong presentation. However, a specimen from private or semi-private (? Note comment on line 224) collection included – Lauer Foundation. The issue here concerns long term access for research and reproducibility. Is the fate of this specimen (and others)

agreed after the lifetime of the present collection, i.e. will it be sold or is there an agreed donation to a fully accredited public museum collection?

Response: The Lauer Foundation for Paleontology, Science and Education (<https://www.lauerfoundationpse.org/>) is an independently operating foundation governed by a corporate Board of Directors. It is registered as a 501(c)(3) nonprofit organization with the U.S. Government and the State of Illinois. The Foundation, rather than any individual, retains ownership of all specimens, adhering to a legal structure similar to that of many public museums. To address specific assurances requested by collaborating institutions, the Foundation's Corporate By-Laws were amended to reinforce its commitment to advancing scientific research and the responsible curation of its fossil collection. These amendments establish the following principles:

- Preservation and Curation – Ensuring the long-term preservation and proper curation of fossils within a climate-controlled indoor facility.
- Scientific and Public Access – Guaranteeing permanent access to the *Fossil Collection* for research, education, and exhibition by the scientific community and accredited museums. This includes public access to type and figured specimens, as well as specimens listed or cited in peer-reviewed publications and other scientifically significant materials.
- Succession Planning – Implementing a structured succession plan to uphold the Foundation's core mission and ensure continuity of its operations.
- Future Ownership and Stewardship – Establishing a framework for the transfer of ownership of the *Fossil Collection* to a public museum or recognized repository, as determined by the Board of Directors, to ensure continued accessibility for scientific study and public benefit.

Furthermore, the Foundation is committed to the perpetual preservation of scientifically significant specimens and will not dispose of any type specimens, figured specimens, or specimens listed or cited in scholarly publications. Any scientifically significant fossil, whether currently held or acquired in the future, will be treated with the same protection as type or figured material.

R#3-6: Figure 2. Clear photographs, but these prompted searches for traces of restored features such as the spiracle (given the completeness of the squamation). Line drawings of best estimates of endoskeletal cartilages (components of mandibular and hyoid arches, pectoral and pelvic skeletons; axial column) would add greatly to the value of this manuscript.

Low angle illumination to complement the UV illuminated details would be a great help. Ideally, it would be good to see skeletal reconstruction, as far as it can be determined.

Response: We have added photographs under normal and UV light, along with interpretative line drawings of both dental and skeletal features, to enhance and clarify our morphological descriptions. While the location of the spiracle could not be determined from the available fossil specimens, it has been inferred in the tentative life reconstruction based on comparisons with extant sharks.

R#3-7: L. 237 – Paired scapulocoracoids fused – this is the kind of detail that deserves support from a line-drawn interpretation of Fig. 2D (for example).

Response: Interpretative line drawings have been implemented as requested.

R#3-8: L. 240+ - Proximal radials (pro, meso, & metapterygium) of the pectoral fin – again, an appeal for line-drawing of the specimen. Adding thick black lines to the photograph is not a good idea (reminiscent of fossils where the boundaries of bones have been painted-in by well-intentioned 19th and early 20th century researchers).

Distal radials? No estimate of minimum number and distribution, even if thought to be incomplete.

Response: We have revised the figures to further clarify and complement our descriptions. Regarding the pectoral radials, we have provided additional information despite their partial preservation:

Furthermore, at least six pectoral radials are discernible in SNSB-BSPG 1878 IV, but due to their obscured nature, no further details can be provided.

R#3-9: L. 244+ Pelvic skeleton – similar issues. How are the fluorescing patches in the photographs interpreted? For example 'the basipterygium appears to have been straight' – without a diagram I'm unsure about the basis for this interpretation.

Response: We have revised the respective figure.

R#3-10: L. 252+. And so on for the dorsal fin skeletons.

Response: We have revised the respective figure.

R#3-11: L. 262. Extremely brief treatment of the axial skeleton. Centra mentioned – but no counts; estimates of numbers per body region, or related to, say, levels of leading edges of fins?

Response: We have expanded the relevant section, which now reads as follows:

The vertebral column is best preserved in JME SOS 4124, which comprises at least 115 cyclospindyl vertebral centra (Figure 4); however, insufficient preservation of the posterior-most centra precludes a more precise count. The transition from mono- to diplospindylous vertebrae occurs above the pectoral (Figure 4B), marked by a distinct decrease in anterior-posterior length between vertebrae 28 and 29 (Figure 4C). A progressive decrease in vertebral length takes place along the caudal fin towards its posterior end. Epaxial (dorsal) elements are discernible, likely representing basidorsals and interdorsals, although a more precise determination is limited due to poor preservation. Hypaxial (ventral) elements could not be observed, but this might be an artefact of preservation.

R#3-12: L. 265. The teeth from Mahlstetten – excellent images, but for the reader new to these sharks, it would be helpful if this plate included an example from the Solnhofen holomorphs – anchoring tooth-morphs to body. What uniquely derived characteristics, or unique combination of general features, are shared?

Response: We believe that it is not necessary to refigure the teeth, as they have already been illustrated in detail in previous works by Thies (2005), Thies & Leidner (2011), and Cappetta (2012).

R#3-13: For tooth terminology, standard, cite Cappetta '87 – or some other source appropriate to task.

Response: As outlined in the M&M section, the dental terminology used largely follows that of Cappetta (2012).

R#3-14: L. 299+ How are the monognathal heterodontal trends known, if complete teeth are only found isolated, from a different locality. Is there a jaw with a more-or-less complete generation of emerged teeth, available for inspection. Is it possible that heterodonty of the kind depicted in Fig. 3 might occur within a single generative tooth set (one family)?

Response: While shark teeth within a single generative tooth set can exhibit considerable ontogenetic variation (as seen, for example, in *Scyliorhinus stellaris*; Berio et al. 2020), we would nevertheless expect to observe a range of size classes, each characterized by a distinct tooth morphotype. Although a complete dentition with well-preserved teeth is not available for †*Bavariscyllium*, the morphological variation observed in the Mahlsetten teeth—including the presence of lateral cusplets that range from rudimentary to more well-defined—closely aligns with that described for the holotype specimen.

Berio, F., Evin, A., Goudemand, N., & Debiais-Thibaud, M. 2020. The intraspecific diversity of tooth morphology in the large-spotted catshark *Scyliorhinus stellaris*: insights into the ontogenetic cues driving sexual dimorphism. *Journal of Anatomy*, 237, 960–978.

R#3-15: L. 306. 'The dermal denticle'. No, there's more than one, and likely more than one kind of crown. Have Thies & Leidner (2011) document dermal denticle variation, sampling from the rostrum and/or leading edges of fins, and compared denticles from dorsal and ventral surfaces of the trunk?

Response: According to Thies & Leidner (2011), the specific regions of the body from which the denticles originate are unknown. We have modified the relevant section, which now reads as follows:

The dermal denticle morphology of †*Bavariscyllium tischlingeri* has been documented by Thies & Leidner (2011), who figured five isolated dermal denticles from the holotype specimen (Thies & Leidner 2011:fig. 44B–E). However, the specific regions of the body from which these denticles originate remain unknown.

R#3-16: L. 314 To assess affinity? 'Affinity' suggests body shapes are being compared to establish galeomorph interrelationships. Surely the question here is to discover whether established clades occupy distinct areas of morphospace and how these align with hypotheses of *Bavariscyllium* affinity. However, the questions (one or more) behind this morphometrics section are not clear. The current text reads as if the morphometrics were done because it's the kind of thing that these sorts of papers include... and then some ideas/observations were written about the graph plots. Comments on the positions of dorsal fins (lines 345-347) don't require morphometric analyses.

Response: We have modified the relevant section, which now reads as follows:

To quantify the body shape of †*Bavariscyllium tischlingeri*, a set of 16 linear body measurements was used (Supplementary Figure S1) to compare this species, along with †*Palaeoscyllium formosum*, to extant demersal sharks with elongate, fusiform precaudal body designs, comprising species in the carcharhiniform families Scyliorhinidae, Atelomycteridae and Pentanchidae, and phenotypically similar sharks in the orectolobiform families Parascylliidae and Hemiscylliidae (Figure 6A; Supplementary Table S1).

R#3-17: L. 367. What would it mean here to be 'allied' with any family? Phenotypic resemblance – signifying what?

Response: We have modified the relevant section, which now reads as follows:

The dataset was further subjected to a linear discriminant analysis (LDA) to explore the separation between the *a priori* groups using the leave-one-out cross-validation (LOOCV) approach. The results indicate that neither †*Bavariscyllium* nor †*Palaeoscyllium* can be phenotypically allied with any of the analyzed families (Supplementary Table S13), with a LOOCV accuracy of 93.2%.

R#3-18: Phylogenetic relationships I've not had time to run the analysis – but pleased to see all relevant materials available in supplementary data. L. 380 'evolutionary steps' – these steps are estimates of character state transitions. For the majority rule consensus tree, what % threshold? (perhaps mentioned in supplementary data – but simple to insert in the figure legend).

Response: We have used a 50% threshold for the majority rule consensus tree and have revised the respective figure caption accordingly.

R#3-19: Fig. 6 – supplementary data states that jackknife and Bremer support values were calculated.

These could be usefully added to Fig. 6A.

Response: Done.

R#3-20: Discussion

L. 425 – Not all orectolobiforms have the interbasal space– see *Cephaloscyllium* -I thought (perhaps mistakenly) that this was the plesiomorphic condition for the group. What might these different kinds of fin skeleton tell us about fin function?

Response: As far as we know, all orectolobiforms have an interbasal space, whereas *Cephaloscyllium* is actually a carcharhiniform, not an orectolobiform.

R#3-21: L. 520+ Final paragraph on body shapes and Jurassic diversification broadens the discussion – for now, this is where the interest value might lie. Perhaps re-order the discussion – move this section before the discussion of tooth shapes and possible diets. And there's always the perennial debate – which comes first – novel feeding apparatus or body shape?

Response: Thank you for the suggestion. We appreciate your perspective on reordering the discussion. However, we would prefer to maintain the current structure, as it preserves a logical flow and helps build a cohesive narrative.

Regarding the classic 'which comes first' debate, we did not intend to address this issue, as doing so properly would require the inclusion of rays, which is beyond the scope of this study. Therefore, we believe it would be premature to tackle this at this stage.

Additional changes

- We have integrated the findings of a recently published study by Dearden et al. (2025) into the discussion, which identifies the Albian–Lutetian shark †*Pararhincodon* as a stem-group member of the Parascylliidae, and the respective section now reads:

The Albian–Lutetian shark †*Pararhincodon* has recently been shown to be a stem-group member of Parascylliidae (Dearden et al. 2025), comprising five species, the oldest of which is †*P. lehmani*, known from an incomplete and poorly preserved articulated specimen from the Cenomanian of Lebanon (Cappetta 1980). Notably, this species also exhibits a single whisker-like throat barbel (Supplementary Figure S7). This feature has previously been overlooked, although it remains possible that, like *Cirrhoscyllium*, †*Pararhincodon* may have had two throat barbels, pending the discovery of more complete fossil material.

Unlike living orectolobiforms and †*Pararhincodon*, which share a curved mesopterygium with a concave trailing edge creating a distinct interbasal space between the meso- and metapterygium (Shirai 1996; Goto 2001; Dearden et al. 2025), the mesopterygium of †*Bavariscyllium* appears to have been straight rather than curved. Extant parascylliids differ from †*Bavariscyllium* and †*Pararhincodon* in having the pro- and mesopterygium fused into a single large cartilage (Goto 2001; Dearden et al. 2025) and well-defined prepelvic processes on the pelvic girdle (Goto 2001), although the latter feature remains unknown in †*Pararhincodon*.

†*Pararhincodon* and extant parascylliids differ from more derived orectolobiforms in having scyliorhinid-like teeth lacking an apron or uvula, similar to †*Bavariscyllium*, but with less well-developed root lobes (Herman et al. 1992; Cappetta 2012). However, the recently described species †*Pararhincodon torquis* from the Upper Cretaceous of the UK provides some evidence of a small labial apron in the recently described species, suggesting that the presence of an apron might represent a plesiomorphic feature rather than a dental trait shared by orectolobiforms more derived than parascylliids (Dearden et al. 2025). In †*Pararhincodon*, the tooth roots are either hemiaulacorhize or holaulacorhize, depending on the species (Cappetta 2012), while in extant parascylliids they always are hemiaulacorhize (Herman et al. 1992).

- We have made a minor correction to the Material and Methods section to clarify the conditions that likely contributed to the exceptional fossil preservation in the Solnhofen Archipelago, which now reads:

The deposits were formed in closely associated depocenters bounded by sponge and occasional coral reefs, some intermittently exposed as small islands. Restricted water circulation and limited water exchange resulted in dysoxic to anoxic bottom conditions that suppressed bioturbation, microbial decay, and scavenging. Combined with rapid burial by abiogenic carbonate precipitation, storm events, and synsedimentary mass flows, these conditions likely facilitated the exceptional fossil preservation (Arratia et al. 2015).

universität
wien

Dr. Sebastian Stumpf
Senior Research Fellow
University of Vienna
Department of Palaeontology
Josef-Holaubek-Platz 2 (UZA II)
1090 Vienna, Austria

Vienna, 1 August 2025

Reviewer #1

The authors have made efforts to address some of the issues highlighted in the initial review. My primary concerns were as follows: the unequivocal attribution of material from a different deposit and age to this species, which is known from skeletons and associated teeth as originally described by Thies (2005); the possibility that the four original skeletons, each measuring less than 25 cm, were more likely juveniles rather than adults, necessitating consideration of potential allometric effects in morphometric analyses; and the challenge of interpreting the coding of morphological characters on specimens, which underpins phylogenetic analyses. While the latter issue has been ameliorated in the revised manuscript, with the inclusion of interesting appendices, the other two concerns remain inadequately addressed. The authors have attempted to justify their first choices without presenting any novel or conclusive arguments.

R#1-1: Additionally, other minor points have not been fully considered, such as the highly unusual presence of a single (throat) barbel in this fossil species, despite evidence suggesting the existence of two. If the authors choose to maintain their first assertion that this fossil shark possesses a single throat barbel—a characteristic not previously observed in any living and fossil species—they should either: consider the possibility that *Bavariscyllium* lacks also paired pelvic fins, as these have not been observed concurrently on the four specimens; or, alternatively, that they must exclude this strange feature (a unique whisker-like barbel as originating from the ventral surface of the throat) from the “amended diagnosis” if its presence

is considered as uncertain, as it is "awaiting substantiation by additional fossil evidence (newly noted).

Response: As outlined in the revised manuscript, we explicitly acknowledge the possibility that †*Bavariscyllum* may have possessed a pair of throat barbels, clearly stating that our interpretation of a single barbel is provisional. Given the available fossil material, we believe it would be inappropriate to assert the presence of a second barbel without direct evidence, just as it would be unwarranted to omit mention of the observed structure altogether. Our cautious wording reflects this uncertainty while describing what is preserved.

As also noted in our initial response, we reiterate that the homology of the observed barbel with those of *Cirrhoscyllum* remains uncertain. A definitive assessment would require unattainable histological data. Furthermore, ongoing research (to be published elsewhere) reveals a greater variability in unbranched barbels among chondrichthyans from the Solnhofen Archipelago than previously recognised. This, alongside intraspecific variation observed in extant actinopterygians (Fox 1999), highlights the interpretative difficulties of such features in extinct taxa.

R#1-2: Concerning the second major disagreement (state of development of skeletons) I don't understand the authors' statement: "Despite the small size of the available specimens and the absence of claspers in any of them, which might suggest juvenile stages, they are interpreted as representing subadult to adult individuals. This interpretation is based on the apparently advanced mineralization of both their endo- and exoskeletons, in conjunction with the prolonged skeletogenesis observed in extant sharks, which starts during late embryonic development and continues well beyond hatching," is unclear to me. The study by López-Romero et al. (2022) effectively illustrates ontogenetic variations in mandible morphology between a carcharhiniform and an orectolobiform. Nonetheless, this research underscores the importance of timing in developmental processes among elasmobranchs and highlights the divergence in shape that occurs even before cartilage differentiation, which is distinct from the issues addressed in the current discussion. It is unequivocal that these specimens are not neonates hatched from eggs; however, as the authors note, there exists a substantial developmental gap between the neonate and adult stages. Their concept of "advanced endo- and exoskeleton mineralization" to justify the adult stage remains unclear to me. López-Romero et al. (2022) only suggested that presence of labial cartilage could be used as evidence of advanced development stage toward adult, I am not sure that these labial cartilage are available or visible in the four skeletons? Claspers, for instance, only manifest in males upon reaching sexual maturity. It is plausible that the specimens in question are juveniles that

have not yet attained sexual maturity. With a supposed adult length of 19-25 cm, *Bavariscyllium* would represent the smallest known species within the Carcharhiniformes/Orectolobiformes clade (currently, *Apristurus* and *Cirrhoscyllium* measure approximately max 30 cm in length), which is significant as these are the earliest representatives of the clade. It is acknowledged that, in terms of overall body dimensions, juveniles are approximately geometrically similar to adults (e.g., Irschick & Hammerschlag 2014). Juveniles are typically characterised by having relatively larger caudal fins than adults, a fact supported by numerous studies (see also Irschick & Hammerschlag 2014), which could potentially affect morphometric outcomes. Upon reviewing the newly available morphometric data, it appears that the four *Bavariscyllium* specimens possess the largest caudal fins (LLL with 28-29% TL or CVM with 31-33% TL) among the 185 living species analyzed. It is important to consider the juvenile state as a potential source of bias in morphometric analysis.

Response: We acknowledge that López-Romero et al. (2022) may not be the most appropriate reference for prolonged skeletogenesis in extant sharks. Accordingly, we have replaced this citation with Summers et al. (2004), which demonstrates that in the extant heterodontiform *Heterodontus francisci*, both the chondro- and viscerocranium remain incompletely mineralised until adulthood. We maintain that our original wording in the revised manuscript remains appropriately cautious. This caution is justified by the relatively small size and absence of claspers in all available specimens, which could suggest immaturity. However, this interpretation is counterbalanced by clear evidence of advanced mineralization, including well-calcified axial and appendicular skeletons.

We respectfully disagree with the reviewer's implication that small body size necessarily indicates immaturity. For instance, holomorphic shark specimens from the Upper Cretaceous of Lebanon, including heterodontiforms (sister group to all other galeomorphs), orectolobiforms, and squaliforms, have been documented with fully developed claspers at body lengths well below 30 cm (Pfeil 2021). These specimens, which are more than 40 million years younger than those from the Solnhofen Archipelago, caution against using small size as a definitive indicator of developmental stage in fossil sharks.

As noted in our initial response, our morphometric analysis provides no evidence for allometric growth in †*Bavariscyllium*, despite specimens ranging from 19 to 25 cm in total length (a maximum size difference of approximately 24%). While this does not entirely rule out allometric growth, conclusive evidence is lacking and awaits the discovery of additional fossil material. This issue is further complicated by the general lack of well-documented ontogenetic series in many extant sharks, particularly among carcharhiniforms with scyliorhinid-like dentitions and parascylliids.

The reviewer is correct that †*Bavariscyllium* exhibits the proportionally longest caudal fin among all shark species examined. However, such outliers are not unprecedented among sharks from the Solnhofen Archipelago. For example, the enigmatic squalomorph †*Protospinax* possesses a body plan that differs markedly from any extant squalomorph (Jambura et al. 2023). As discussed in the manuscript, †*Palaeoscyllium* exhibits a scyliorhinid-like dentition, yet differs not only from †*Bavariscyllium* but also from all modern sharks with comparable tooth morphology. These examples highlight that body plans divergent from those of extant sharks do not necessarily reflect immature stages, but may instead represent lineage-specific specializations.

In summary, we consider our interpretation of the available †*Bavariscyllium* specimens as subadult to adult individuals to be well supported. We have, nonetheless, taken care to reflect the uncertainties inherent in working with fossil sharks, and we believe that the revised manuscript appropriately conveys a balanced and cautious perspective.

R#1-3: Concerning the first major disagreement (attribution of Kimmeridgian teeth to these Tithonian specimens) the authors do not address the question of whether these teeth can be definitively attributed to this species. I remain unconvinced of such attribution as they remain also convinced of this proximity: " While we were already confident in our conclusion prior to obtaining these images, the new findings now provide definitive support for our assertion that the Mahlstetten teeth can be unambiguously referred to †*Bavariscyllium tischlinger*". Despite efforts to newly figure teeth of the analysed skeletons, the question of whether the Kimmeridgian teeth are identical to the holotype described by Thies (2005) remains unresolved for me. Although the overall morphology of the new figuration (Figure 2C-D) of the "two" teeth is partially identifiable for meaningful morphological assessment, certain features are clearly absent. For instance, neither two lateral denticles nor pronounced ornamentation of the lingual face (as visible in most of the Kimmeridgian teeth) are observed in Thies' material (2005) or in the new Figure 2C-D. The inaccessibility of the holotype teeth's morphology due to the consolidant is indeed unfortunate, as it could have facilitated confirmation or refinement of the match. Regrettably, even with this new representation, it remains challenging to classify the Kimmeridgian material within this species, and even more so to be used as support of the new 'amended diagnosis' of this taxon, which is formally based on a Tithonian holotype with already figured teeth.

Currently, the Kimmeridgian teeth could also be attributable to †*Palaeoscyllium tenuidens* from the Bathonian of England (Underwood & Ward 2004), to †*P. formosum* from the Kimmeridgian-Tithonian of Germany, France, and England (Candoni 1993; Underwood

2002; Kriwet & Klug 2004), or even to †*Thiesus concavus* from the Valanginian of France (Guinot et al. 2014), as they further elaborate. The authors propose a convoluted process of synonymy, wherein indeterminate isolated teeth are assigned to a known species, subsequently leading to the synonymization of other, more recent species based on isolated teeth. To substantiate my uncertainty, I also reviewed that Thies (2005) has further identified Kimmeridgian teeth from another deposit in Germany, attributing them to the genus *Bavariscyllium*, though he refrains from assigning them to the sole species within the genus. These teeth exhibit a closer resemblance to the holotype than to those illustrated by the authors (not discussed in this manuscript, but included in the species reference list). I have included a figure compiled for this purpose, featuring the teeth of the *Bavariscyllium* holotype, those from Kimmeridgian and newly attributed to the same species, those assigned to the *Bavariscyllium* (sp.) by Thies (2005) from Kimmeridgian also, as well as those from other contemporary and/or synonymized genera as identified by the authors. At present, the task of associating nearly all of these teeth (excepted that of *Palaeoscyllium*) with a single species *Bavariscyllium tischlingeri* presents a challenge that remains somewhat elusive.

If the authors wish to code their reconstructed dentition from Kimmeridgian of Mahlsetten as a character of *Bavariscyllium* for phylogeny, they may do so as material related to this genus and as an element of interest, albeit with a reservation regarding their systematic attribution (similarly with *Thiesus concavus*, which they consider a synonym), which would not fundamentally alter the results of the phylogenetic and morphometric study. However, it would be more appropriate not to include the teeth in the diagnosis, given the considerable uncertainty regarding their taxonomic affiliation. The diversity of Jurassic elasmobranchs, particularly Carcharhiniformes, is limited and complex, as the material is fragmentary and morphologically not very diverse (mainly belonging to scyliorhinid-like forms). It would be regrettable to consolidate them all around a few known taxa. Reviewer3 suggested including the teeth of the holotype. As the authors rightly note, these teeth are well known and have been illustrated in detail in previous works by Thies (2005), Thies & Leidner (2011), and Cappetta (2012). However, I believe it would be beneficial for the reader to compare them with the Kimmeridgian teeth attributed to this taxon by authors.

Response: We acknowledge the reviewer's concerns regarding the attribution of the Mahlsetten teeth to †*Bavariscyllium tischlingeri*, particularly in light of the incompleteness and poor preservation of the holotype's dentition. However, we respectfully maintain that the dental characters documented in the Mahlsetten material provide a coherent basis for referring these teeth to †*B. tischlingeri*.

As outlined in our revised manuscript, the labial tooth ornamentation in †*Bavariscyllium*, visible in both the holotype and in the newly attributed specimens from

Mahlstetten, consists of two oblique ridges from which weaker vertical ridges may branch off. This pattern is also found in †*Thiesus concavus* from the Valanginian of France, but differs markedly from the prominent labial ornamentation in †*Palaeoscyllium*, which displays numerous strong, vertically oriented ridges (e.g., Underwood 2002; Kriwet & Klug 2004; Underwood & Ward 2004).

While we agree that the teeth described by Thies (2005) are fragmentary and limit direct comparisons, our study contributes new information by identifying previously overlooked dental features in the holotype specimen. These newly recognised characters are consistent with the Mahlstetten teeth, including the variation of lateral cusplets, which range from rudimentary to more pronounced forms. Such a degree of variability is commonly observed in extant elasmobranchs and often linked to ontogenetic or gynandric heterodonty, particularly in carcharhiniforms with generalised clutching-type dentitions.

Regarding the potential attribution to †*Thiesus concavus*, we acknowledge that the morphological range observed in the Mahlstetten teeth includes features previously considered diagnostic for that taxon. However, our intention is not to propose a “convoluted synonymy” based on isolated teeth alone, but rather to emphasise the limitations inherent in taxonomy based solely on dental remains. In this context, we interpret the observed variability as falling within the expected intraspecific range of †*Bavariscyllum tischlingeri*, a view that also encompasses the material from the Kimmeridgian of northern Germany. These considerations are now reflected in the amended manuscript.

In summary, while a degree of uncertainty is unavoidable when working with isolated fossil teeth, we consider the referral of the Mahlstetten teeth to †*Bavariscyllum tischlingeri* to be both plausible and well supported by the available evidence. We have modified the relevant section, which now reads:

Discussion

[...]

The teeth from the lower Kimmeridgian of Mahlstetten, which share characteristic dental features with the holotype of †*B. tischlingeri* (including a concave labial crown edge and a pair of labially displaced lateral heels, each with small cusplets and typically an oblique labial ridge, as well as divergent labial root lobes and a weak lingual protuberance pierced basally by a single foramen), provide valuable new insights into the species’ dental morphology. These scyliorhinid-like teeth, characterised by a low hemiaulacorhize root exhibiting strongly flared lobes, closely resemble †*Palaeoscyllium tenuidens* from the Bathonian of England (Underwood & Ward 2004) and †*P. formosum*

from the Kimmeridgian–Tithonian of Germany, France, and England (Candoni 1993; Underwood 2002; Kriwet & Klug 2004). The closest similarities are with †*Thiesus concavus* from the Valanginian of France (Guinot et al. 2014), with some teeth being nearly indistinguishable, differing from †*T. concavus* only in having main cusps that are slightly more robust and less lingually bent. However, this very minor difference, along with the shared lack of labial ornamentation in some teeth, is more likely to reflect intra- rather than interspecific variation, such as gynandric and/or ontogenetic heterodonty, as documented in many extant elasmobranch species (e.g., Gutteridge & Bennett 2014; Underwood et al. 2015; Cullen & Marshall 2019; Ebersole et al. 2023) and particularly well expressed in carcharhiniforms with generalised clutching-type dentitions (Herman et al. 1990; Soares & de Carvalho 2019; Berio et al. 2020, 2022). Consequently, due to the striking similarities between the Mahlsetten teeth and †*T. concavus*, we propose treating the latter as a junior synonym of †*B. tischlingeri*, extending its stratigraphic range into the Early Cretaceous. The two fragmentary teeth from the Kimmeridgian of northern Germany described by Thies (2005), originally referred to †*Bavariscyllium* sp. and both displaying the characteristic oblique labial ridges, are likewise referred here to †*B. tischlingeri*. Additionally, †*Bavariscyllium* may also extend down into the Middle Jurassic, as suggested by a lateral tooth from the Bathonian of England, initially reported by Underwood & Ward (2004) as Scyliorhinidae? gen. indet. and later considered by Guinot et al. (2014) as potentially representing an unnamed species of †*Thiesus*, though this cannot be confirmed until more dental material is found.

R#1-4: I also have other disagreements with the authors' response regarding the use of backbones in phylogeny (e.g. "in phylogenetic studies, the use of topological constraints may obscure phylogenetic uncertainty within the dataset."). This issue is particularly pertinent when integrating modern taxa with an ancient fossil ant, as employing topological constraints is arguably the only effective way to merge robust neontological (molecular) data with the current phylogenetic framework. Although it is recognized that this method may polarize certain morphological character states a posteriori, it remains a persuasive approach. However, since the phylogenetic resolution of fossil taxa, as presented by the authors, remains unclear, I see no reason to oppose preserving this ambiguity.

Response: While we acknowledge that topological constraints can be valuable for integrating molecular and fossil data, the very high proportion of missing data (>50%) in †*Bavariscyllium* (and †*Palaeoscyllium*) limits the efficacy of this approach in the present case. In this context, enforcing a backbone risks overinterpreting sparse data

and may create a misleading impression of phylogenetic resolution. As our primary goal is to explore morphological character-based relationships without imposing preconceived placements, preserving ambiguity more accurately reflects the limitations of the available data and helps to avoid interpretive bias.

We would welcome any suggestions of specific synapomorphic features that could support a more constrained placement of these fossil taxa. However, in the absence of such evidence, we see no compelling reason to override the uncertainty that emerges from the current analysis. Accordingly, the reviewer's reluctance to accept a result that explicitly reflects this uncertainty remains difficult to understand. Moreover, no arguments have been offered regarding the potential benefits of achieving a more resolved phylogeny through topological constraints, particularly in the case of fossil taxa with uncertain affinities under a parsimony-based framework, which aims to explain observed similarities through common ancestry.

Importantly, our results do not challenge established phylogenetic groups. Rather, they reflect the limited resolution afforded by the available data and the uncertainty in the placement of the taxa of interest relative to those groups. In our view, explicitly acknowledging this uncertainty is essential for a realistic and methodologically transparent understanding of their evolutionary history and relationships.

References

- Fox, H. 1999. Barbels and barbel-like tentacular structures in sub-mammalian vertebrates: a review. *Hydrobiologia* 403, 153–193.
- Jambura, P.L., Villalobos-Segura, E., Türtscher, J., Begat, A., Staggl, M.A., Stumpf, S., Kindlimann, R., Klug, S., Lacombat, F., Pohl, B., Maisey, J.G., Naylor, G.J.P., Kriwet, J. 2023. Systematics and phylogenetic interrelationships of the enigmatic Late Jurassic shark *Protospinax annectans* Woodward, 1918 with comments on the shark–ray sister group relationship. *Diversity*, 15, 311.
- Kriwet, J., Klug, S. 2004. Late Jurassic selachians (Chondrichthyes, Elasmobranchii) from southern Germany: re-evaluation on taxonomy and diversity. *Zitteliana, Reihe A*, 44, 67–95.
- Pfeil, F.H. 2021. The new family Mesiteiidae (Chondrichthyes, Orectolobiformes), based on *Mesiteia emiliae* Kramberger, 1884. A contribution to the Upper Cretaceous (early Cenomanian) shark fauna from Lebanon. In: Pradel, A., Denton J.S.S., Janvier, P. (eds.), *Ancient Fishes and their Living Relatives: a Tribute to John G. Maisey*. Verlag Dr. Friedrich Pfeil, Munich, pp. 102–182.

- Summers, A.P., Ketcham, R.A., Rowe, T. 2004. Structure and function of the horn shark (*Heterodontus francisci*) cranium through ontogeny: Development of a hard prey specialist. *Journal of Morphology*, 260, 1–12.
- Underwood, C.J. 2002. Sharks, rays and a chimaeroid from the Kimmeridgian (Late Jurassic) of Ringstead, southern England. *Palaeontology*, 45, 297–325.
- Underwood, C.J. & Ward, D.J. 2004. Neoselachian sharks and rays from the British Bathonian (Middle Jurassic). *Palaeontology*, 473, 447–501.